# Overfitting Can Be Harmless for Basis Pursuit, But Only to a Degree

**Peizhong Ju**
School of ECE
Purdue University
West Lafayette, IN 47906
jup@purdue.edu

**Xiaojun Lin**
School of ECE
Purdue University
West Lafayette, IN 47906
linx@purdue.edu

**Jia Liu**
Department of ECE
The Ohio State University
Columbus, OH 43210
liu@ece.osu.edu

## Abstract

Recently, there have been significant interests in studying the so-called "double-descent" of the generalization error of linear regression models under the overparameterized and overfitting regime, with the hope that such analysis may provide the first step towards understanding why overparameterized deep neural networks (DNN) still generalize well. However, to date most of these studies focused on the min $\ell_2$-norm solution that overfits the data. In contrast, in this paper we study the overfitting solution that minimizes the $\ell_1$-norm, which is known as Basis Pursuit (BP) in the compressed sensing literature. Under a sparse true linear regression model with $p$ *i.i.d.* Gaussian features, we show that for a large range of $p$ up to a limit that grows exponentially with the number of samples $n$, with high probability the model error of BP is upper bounded by a value that decreases with $p$. To the best of our knowledge, this is the first analytical result in the literature establishing the double-descent of overfitting BP for finite $n$ and $p$. Further, our results reveal significant differences between the double-descent of BP and min $\ell_2$-norm solutions. Specifically, the double-descent upper-bound of BP is independent of the signal strength, and for high SNR and sparse models the descent-floor of BP can be much lower and wider than that of min $\ell_2$-norm solutions.

## 1 Introduction

One of the mysteries of deep neural networks (DNN) is that they are not only heavily overparameterized so that they can fit the training data (even perturbed with noise) nearly perfectly [3, 4, 14, 32], but still produce models that generalize well to new test data [2, 36]. This combination of overfitting and good generalization challenges the classical wisdom in statistical learning (e.g., the well-known bias-variance tradeoff) [10, 20, 21, 23, 31, 34]. As a first step towards understanding why overparameterization and overfitting may be harmless, a line of recent work has focused on linear regression models [6, 7, 8, 19, 25, 29]. Indeed, such results have demonstrated an interesting "double descent" phenomenon for linear models. Roughly speaking, let $n$ be the number of training samples, and $p$ be the number of parameters of a linear regression model. As $p$ approaches $n$ from below, the test error of the model (that tries to best fit the training data) first decreases and then increases to infinity, which is consistent with the well-understood bias-variance trade-off. As $p$ further increases beyond $n$, overfitting starts to occur (i.e., the training error will always be zero). However, if one chooses the overfitting solution that minimizes the $\ell_2$-norm, the test error decreases again as $p$ increases further. Such observations, although for models very different from DNN, provide some hint why overfitting solutions may still generalize well.

To date most studies along this direction have focused on the minimum $\ell_2$-norm overfitting solutions [7, 19, 25, 28, 29]. One possible motivation is that, at least for linear regression problems, Stochastic

Gradient Descent (SGD), which is often used to train DNNs, is believed to produce the min $\ell_2$-norm overfitting solutions [36]. However, it is unclear whether SGD will still produce the min $\ell_2$-norm overfitting solutions for more general models, such as DNN. Therefore, it is important to understand whether and how double-descent occurs for other types of overfitting solutions. Further, the min $\ell_2$-norm solution usually does not promote sparsity. Instead, it tends to yield small weights spread across nearly all features, which leads to distinct characteristics of its double-descent curve (see further comparisons below). The double-descent of other overfitting solutions will likely have different characteristics. By understanding these differences, we may be able to discern which type of overfitting solutions may approximate the generalization power of DNN better.

In this paper, we focus on the overfitting solution with the minimum $\ell_1$-norm. This is known as Basis Pursuit (BP) in the compressed sensing literature [12, 13]. There are several reasons why we are interested in BP with overfitting. First, similar to $\ell_2$-minimization, it does not involve any explicit regularization parameters, and thus can be used even if we do not know the sparsity level or the noise level. Second, it is well-known that using $\ell_1$-norm promotes sparse solutions [9, 12, 16, 27, 33, 37], which is useful in the overparameterized regime. Third, it is known that the $\ell_1$-norm of the model is closely related to its "fat-shattering dimension," which is also related to the Vapnik-Chervonenkis (V-C) dimension and the model capacity [5]. Thus, BP seems to have the appealing flavor of "Occam's razor" [11], i.e., to use the simplest explanation that matches the training data. However, until now the double descent of BP has not been well studied. The numerical results in [29] suggest that, for a wide range of $p$, BP indeed exhibits double-descent and produces low test-errors. However, no analysis is provided in [29]. In the compressed sensing literature, test-error bounds for BP were provided for the overparameterized regime, see, e.g., [16]. However, the notion of BP therein is different as it requires that the model does *not* overfit the training data. Hence, such results cannot be used to explain the "double-descent" of BP in the overfitting regime. For classification problems, minimum $\ell_1$-norm solutions that interpolate categorical data were studied in [24]. However, the notion of "overfitting" in [24] for classification problems is quite different from that for the regression problems studied in this paper and [29]. To the best of our knowledge, the only work that analyzed the double-descent of overfitting BP in a regression setting is [28]. However, this work mainly studies the setting where both $n$ and $p$ grow to infinity at a fixed ratio. As we show later, BP exhibits interesting dynamics of double-descent when $p$ is exponentially larger than $n$, which unfortunately collapsed into a single point of $p/n \to \infty$ in the setting of [28]. In summary, a thorough study of the double-descent of BP for finite $n$ and $p$ is still missing.

The main contribution of this paper is thus to provide new analytical bounds on the model error of BP in the overparameterized regime. As in [7], we consider a simple linear regression model with $p$ *i.i.d.* Gaussian features. We assume that the true model is sparse, and the sparsity level is $s$. BP is used to train the model by exactly fitting $n$ training samples. For a range of $p$ up to a value that grows exponentially with $n$, we show an upper bound on the model error that decreases with $p$, which explains the "double descent" phenomenon observed for BP in the numerical results in [29]. To the best of our knowledge, this is the first analytical result in the literature establishing the double-descent of min $\ell_1$-norm overfitting solutions for finite $n$ and $p$.

Our results reveal significant differences between the double-descent of BP and min $\ell_2$-norm solutions. First, our upper bound for the model error of BP is independent of the signal strength (i.e., $\|\beta\|_2$ in the model in Eq. (5)). In contrast, the double descent of min $\ell_2$-norm overfitting solutions usually increases with $\|\beta\|_2$, suggesting that some signals are "spilled" into the model error. Second, the double descent of BP is much slower (polynomial in $\log p$) than that of min $\ell_2$-norm overfitting solutions (polynomial in $p$). On the other hand, this also means that the double-descent of BP manifests over a larger range of $p$ and is easier to observe than that of min $\ell_2$-norm overfitting solutions. Third, with both $\ell_1$-norm and $\ell_2$-norm minimization, there is a "descent floor" where the model error reaches the lowest level. However, at high signal-to-noise ratio (SNR) and for sparse models, the descent floor of BP is both lower (by a factor proportional to $\sqrt{\frac{1}{s}\sqrt{\text{SNR}}}$) and wider (for a range of $p$ exponential in $n$) than that of min $\ell_2$-norm solutions.

One additional insight revealed by our proof is the connection between the model error of BP and the ability for an overparameterized model to fit *only* the noise. Roughly speaking, as long as the model is able to fit only the noise with small $\ell_1$-norm solutions, BP will also produce small model errors (see Section 4). This behavior also appears to be unique to BP.

Finally, our results also reveal certain limitations of overfitting, i.e., the descent floor of either BP or min $\ell_2$-norm solutions cannot be as low as regularized (and non-overfitting) solutions such as LASSO [33]. However, the type of explicit regularization used in LASSO is usually not used in DNNs. Thus, it remains an open question how to find practical overfitting solutions that can achieve even lower generalization errors without any explicit regularization.

## 2   Problem setting

Consider a linear model as follows:

$$y = x^T \underline{\beta} + \epsilon, \tag{1}$$

where $x \in \mathbb{R}^p$ is a vector of $p$ features, $y \in \mathbb{R}$ denotes the output, $\epsilon \in \mathbb{R}$ denotes the noise, and $\underline{\beta} \in \mathbb{R}^p$ denotes the regressor vector. We assume that each element of $x$ follows *i.i.d.* standard Gaussian distribution, and $\epsilon$ follows independent Gaussian distribution with zero mean and variance $\sigma^2$. Let $s$ denote the sparsity of $\underline{\beta}$, i.e., $\underline{\beta}$ has at most $s$ non-zero elements. Without loss of generality, we assume that all non-zero elements of $\underline{\beta}$ are in the first $s$ elements. For any $p \times 1$ vector $\alpha$ (such as $\underline{\beta}$), we use $\alpha[i]$ to denote its $i$-th element, use $\alpha_0$ to denote the $s \times 1$ vector that consists of the first $s$ elements of $\alpha$, and use $\alpha_1$ to denote the $(p - s) \times 1$ vector that consists of the remaining elements of $\alpha$. With this notation, we have $\underline{\beta} = \begin{bmatrix} \underline{\beta}_0 \\ \mathbf{0} \end{bmatrix}$.

Let $\underline{\beta}$ be the true regressor and let $\hat{\underline{\beta}}$ be an estimate of $\underline{\beta}$ obtained from the training data. Let $\underline{w} := \hat{\underline{\beta}} - \underline{\beta}$. According to our model setting, the expected test error satisfies

$$\mathbb{E}_{x,\epsilon} \left[ \left( x^T \hat{\underline{\beta}} - (x^T \underline{\beta} + \epsilon) \right)^2 \right] = \mathbb{E}_{x,\epsilon} \left[ (x^T \underline{w} - \epsilon)^2 \right] = \|\underline{w}\|_2^2 + \sigma^2. \tag{2}$$

Since $\sigma^2$ is given, in the rest of the paper we will mostly focus on the model error $\|\underline{w}\|_2$. Note that if $\hat{\underline{\beta}} = 0$, we have $\|\underline{w}\|_2^2 = \|\underline{\beta}\|_2^2$, which is also the average strength of the signal $x^T \underline{\beta}$ and is referred to as the "null risk" in [19]. We define the signal-to-noise ratio as $\text{SNR} := \|\underline{\beta}\|_2^2 / \sigma^2$.

We next describe how BP computes $\hat{\underline{\beta}}$ from training data $(\mathbf{X}_{\text{train}}, \mathbf{Y}_{\text{train}})$, where $\mathbf{X}_{\text{train}} \in \mathbb{R}^{n \times p}$ and $\mathbf{Y}_{\text{train}} \in \mathbb{R}^n$. For ease of analysis, we normalize each column of $\mathbf{X}_{\text{train}}$ as follows. Assume that we have $n$ *i.i.d.* samples in the form of Eq. (1). For each sample $k$, we first divide both sides of Eq. (1) by $\sqrt{n}$, i.e.,

$$\frac{y_k}{\sqrt{n}} = \left( \frac{x_k}{\sqrt{n}} \right)^T \underline{\beta} + \frac{\epsilon_k}{\sqrt{n}}. \tag{3}$$

We then form a matrix $\mathbf{H} \in \mathbb{R}^{n \times p}$ so that each row $k$ is the sample $x_k^T$. Writing $\mathbf{H} = [\mathbf{H}_1 \ \mathbf{H}_2 \ \cdots \ \mathbf{H}_p]$, we then have $\mathbb{E}[\|\mathbf{H}_i\|_2^2] = n$, for each column $i \in \{1, 2, \cdots, p\}$. Now, let $\mathbf{X}_{\text{train}} = [\mathbf{X}_1 \ \mathbf{X}_2 \ \cdots \ \mathbf{X}_p]$ be normalized in such a way that

$$\mathbf{X}_i = \frac{\mathbf{H}_i}{\|\mathbf{H}_i\|_2}, \text{ for all } i \in \{1, 2, \cdots, p\}, \tag{4}$$

and let each row $k$ of $\mathbf{Y}_{\text{train}}$ and $\epsilon_{\text{train}}$ be the corresponding values of $y_k / \sqrt{n}$ and $\epsilon_k / \sqrt{n}$ of the sample. Then, each column $\mathbf{X}_i$ will have a unit $\ell_2$-norm. We can then write the training data as

$$\mathbf{Y}_{\text{train}} = \mathbf{X}_{\text{train}} \beta + \epsilon_{\text{train}}, \tag{5}$$

where

$$\beta[i] = \frac{\|\mathbf{H}_i\|_2}{\sqrt{n}} \underline{\beta}[i] \text{ for all } i \in \{1, 2, \cdots, p\}. \tag{6}$$

Note that the above normalization of $\mathbf{X}_{\text{train}}$ leads to a small distortion of the ground truth from $\underline{\beta}$ to $\beta$, but it eases our subsequent analysis. When $n$ is large, the distortion is small, which will be made precise below (in Lemma 1). Further, we have $\mathbb{E}[\|\epsilon_{\text{train}}\|_2^2] = \sigma^2$.

In the rest of this paper, we focus on the situation of overparameterization, i.e., $p > n$. Among many different estimators of $\beta$, we are interested in those that perfectly fit the training data, i.e.,

$$\mathbf{X}_{\text{train}}\hat{\beta} = \mathbf{Y}_{\text{train}}. \tag{7}$$

When $p > n$, there are infinitely many $\hat{\beta}$'s that satisfy Eq. (7). In BP [13], $\hat{\beta}$ is chosen by solving the following problem

$$\min_{\tilde{\beta}} \|\tilde{\beta}\|_1, \ \ \text{subject to } \mathbf{X}_{\text{train}}\tilde{\beta} = \mathbf{Y}_{\text{train}}. \tag{8}$$

In other words, given $\mathbf{X}_{\text{train}}$ and $\mathbf{Y}_{\text{train}}$, BP finds the overfitting solution with the minimal $\ell_1$-norm. Note that as long as $\mathbf{X}_{\text{train}}$ has full row-rank (which occurs almost surely), Eq. (8) always has a solution. Further, when Eq. (8) has one or multiple solutions, there must exist one with at most $n$ non-zero elements (we prove this fact as Lemma 10 in Appendix A of Supplementary Material). We thus use $\hat{\beta}^{\text{BP}}$ to denote any such solution with at most $n$ non-zero elements. Define $w^{\text{BP}} := \hat{\beta}^{\text{BP}} - \beta$. In the rest of our paper, we will show how to estimate the model error $\|w^{\text{BP}}\|_2$ of BP as a function of the system parameters such as $n$, $p$, $s$, and $\sigma^2$. Note that before we apply the solution of BP in Eq. (8) to new test data, we should re-scale it back to (see Eq. (6))

$$\hat{\underline{\beta}}^{\text{BP}}[i] = \frac{\sqrt{n}\hat{\beta}^{\text{BP}}[i]}{\|\mathbf{H}_i\|_2}, \ \text{for all } i \in \{1, 2, \cdots, p\},$$

and measure the generalization performance by the unscaled version of $w^{\text{BP}}$, i.e., $\underline{w}^{\text{BP}} = \hat{\underline{\beta}}^{\text{BP}} - \beta$. The following lemma shows that, when $n$ is large, the difference between $w^{\text{BP}}$ and $\underline{w}^{\text{BP}}$ in term of either $\ell_1$-norm or $\ell_2$-norm is within a factor of $\sqrt{2}$ with high probability.

**Lemma 1.** *For both $d = 1$ and $d = 2$, we have*

$$\Pr\left(\left\{\|\underline{w}^{BP}\|_d \leq \sqrt{2}\|w^{BP}\|_d\right\}\right) \geq 1 - \exp\left(-\frac{n}{16} + \ln(2n)\right),$$

$$\Pr\left(\left\{\|w^{BP}\|_d \leq \sqrt{2}\|\underline{w}^{BP}\|_d\right\}\right) \geq 1 - \exp\left(-\frac{2 - \sqrt{3}}{2}n + \ln(2n)\right).$$

Therefore, in the rest of the paper we will focus on bounding $\|w^{\text{BP}}\|_2$.

Note that the model error of BP was also studied in [16]. However, the notion of BP therein is different in that the estimator $\hat{\beta}$ only needs to satisfy $\|\mathbf{Y}_{\text{train}} - \mathbf{X}_{\text{train}}\hat{\beta}\|_2 \leq \delta$. The main result (Theorem 3.1) of [16] requires $\delta$ to be greater than the noise level $\|\epsilon_{\text{train}}\|_2$, and thus cannot be zero. (An earlier version [15] of [16] incorrectly claimed that $\delta$ can be 0 when $\|\epsilon_{\text{train}}\|_2 > 0$, which is later corrected in [16].) Therefore, the result of [16] does not capture the performance of BP for the overfitting setting of Eq. (7). Similarly, the analysis of BP in [17] assumes no observation noise, which is also different from Eq. (7).

## 3 Main results

Our main result is the following upper bound on the model error of BP with overfitting.

**Theorem 2** (Upper Bound on $\|w^{\text{BP}}\|_2$). *When $s \leq \sqrt{\frac{n}{7168\ln(16n)}}$, if $p \in \left[(16n)^4, \ \exp\left(\frac{n}{1792s^2}\right)\right]$, then with probability at least $1 - 6/p$, we have*

$$\frac{\|w^{BP}\|_2}{\|\epsilon_{train}\|_2} \leq 2 + 8\left(\frac{7n}{\ln p}\right)^{1/4}. \tag{9}$$

It is well known that the model error of the minimum MSE (mean-square-error) solution has a peak when $p$ approaches $n$ from below [7]. Further, when $p = n$, $\hat{\beta}^{\text{BP}}$ coincides with the min-MSE solution with high probability. Thus, Theorem 2 shows that the model error of overfitting BP must decrease from that peak (i.e., may exhibit double descent) when $p$ increases beyond $n$, up to a value exponential in $n$. Note that the assumption $s \leq \sqrt{\frac{n}{7168\ln(16n)}}$, which states that the true model is

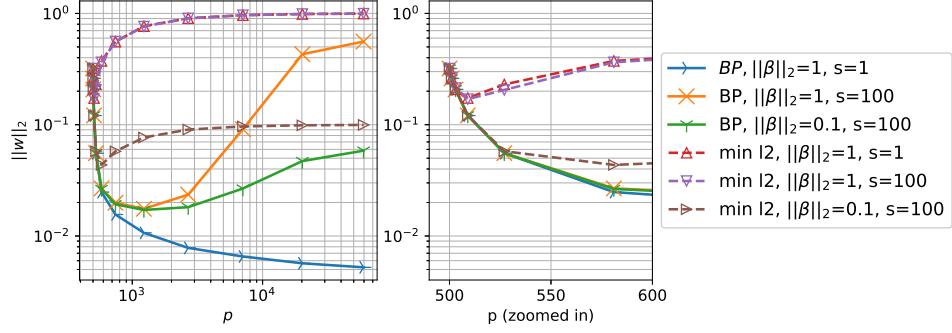

Figure 1: Compare BP with min $\ell_2$-norm for different values of $\|\beta\|_2$ and $s$, where $\|\epsilon_{\text{train}}\|_2 = 0.01$, $n = 500$. The right figure is a zoomed-in version of the left figure for $p \in [500, 600]$.

sufficiently sparse[1], implies that the interval $\left[(16n)^4,\ \exp\left(\frac{n}{1792 s^2}\right)\right]$ is not empty. We note that the constants in Theorem 2 may be loose (e.g., the values of $n$ and $p$ need to be quite large for $p$ to fall into the above interval). These large constants are partly due to our goal to obtain high-probability results, and they could be further optimized. Nonetheless, our numerical results below suggest that the predicted trends (e.g., for double descent) hold for much smaller $n$ and $p$.

The upper limit of $p$ in Theorem 2 suggests a descent floor for BP, which is stated below. Please refer to Supplementary Material (Appendix G) for the proof.

**Corollary 3.** *If* $1 \le s \le \sqrt{\frac{n}{7168 \ln(16n)}}$, *then by setting* $p = \left\lfloor \exp\left(\frac{n}{1792 s^2}\right)\right\rfloor$, *we have*

$$\frac{\|w^{BP}\|_2}{\|\epsilon_{train}\|_2} \le 2 + 32\sqrt{14}\sqrt{s} \tag{10}$$

*with probability at least* $1 - 6/p$.

To the best of our knowledge, Theorem 2 and Corollary 3 are the first in the literature to quantify the double-descent of overfitting BP for finite $n$ and $p$. Although these results mainly focus on large $p$, they reveal several important insights, highlighting the significant differences (despite some similarity) between the double-descent of BP and min $\ell_2$-norm solutions. (The codes for the following numerical experiments can be found in our Github page[2].)

**(i) The double-descent upper-bound of BP is independent of $\|\beta\|_2$. In contrast, the descent of min $\ell_2$-norm solutions is raised by $\|\beta\|_2$.** The upper bounds in both Theorem 2 and Corollary 3 do not depend on the signal strength $\|\beta\|_2$. For min $\ell_2$-norm overfitting solutions, however, if we let $\underline{w}^{\ell_2}$ denote the model error, for comparable Gaussian models we can obtain (see, e.g., Theorem 2 of [7]):

$$\mathbb{E}[\|\underline{w}^{\ell_2}\|_2^2] = \|\beta\|_2^2 \left(1 - \frac{n}{p}\right) + \frac{\sigma^2 n}{p - n - 1}, \text{ for } p \ge n + 2. \tag{11}$$

To compare with Eq. (9), recall that $\mathbb{E}[\|\epsilon_{\text{train}}\|_2^2] = \sigma^2$. (In fact, we can show that, when $n$ is large, $\|\epsilon_{\text{train}}\|_2$ is close to $\sigma$ with high probability. See Supplementary Material (Appendix B).) From Eq. (11), we can see that the right-hand-side increases with $\|\beta\|_2$. This difference between Eq. (9) and Eq. (11) is confirmed by our numerical results in Fig. 1[3]: When we compare $\|\beta\|_2 = 1$ and $\|\beta\|_2 = 0.1$ (both with $s = 100$), the descent of min $\ell_2$-norm solutions (dashed $\triangledown$ and $\triangleright$) varies significantly with $\|\beta\|_2$, while the descent of BP (solid $\times$ and $\curlyvee$) does not vary much with $\|\beta\|_2$ within the region where the model error decreases with $p$. Intuitively, when the signal strength is strong, it should be easier to detect the true model. The deterioration with $\|\beta\|_2$ suggests that some of the signal is "spilled" into the model error of min $\ell_2$-norm solutions, which does not occur for BP in the descent region.

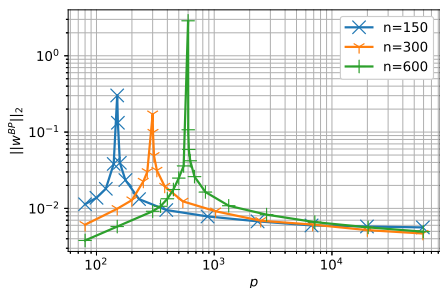
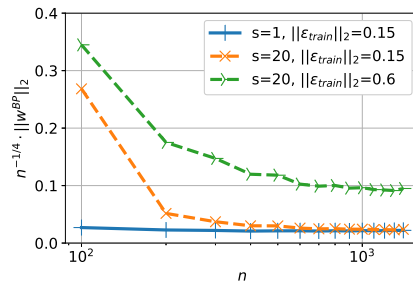

Figure 2: Curves of $\|w^{\text{BP}}\|_2$ with different $n$, where $\|\beta\|_2 = 1$, $\|\epsilon_{\text{train}}\|_2 = 0.01$, $s = 1$. (Note that for $p < n$ we report the model error of the min-MSE solutions.)

Figure 3: Curves of $n^{-1/4}\|w^{\text{BP}}\|_2$, where $p = 5000$, $\|\beta\|_2 = 1$.

*Remark:* Although the term "double descent" is often used in the literature [7], as one can see from Fig. 1, the model error will eventually increase again when $p$ is very large. For BP, the reason is that, when $p$ is very large, there will be some columns of $\mathbf{X}_{\text{train}}$ very similar to those of true features. Then, BP will pick those very similar but wrong features, and the error will approach the null risk. As a result, the whole double-descent behavior becomes "descent-ascent($p < n$)-descent($p > n$)-ascent". Note that similar behavior also arises for the model error of min $\ell_2$-norm overfitting solutions (see Fig. 1). Throughout this paper, we will use the term "descent region" to refer to the second descent part ($p > n$), before the model error ascends for much larger $p$.

**(ii) The descent of BP is slower but easier to observe.** The right-hand-side of Eq. (9) is inversely proportional to $\ln^{1/4} p$, which is slower than the inverse-in-$p$ descent in Eq. (11) for min $\ell_2$-norm solutions. This slower descent can again be verified by the numerical result in Fig. 1. On the other hand, this property also means that the descent of BP is much easier to observe as it holds for an exponentially-large range of $p$, which is in sharp contrast to min $\ell_2$-norm solutions, whose descent quickly stops and bounces back to the null risk (see Fig. 1).

Readers may ask whether the slow descent in $\ln p$ is fundamental to BP, or is just because Theorem 2 is an upper bound. In fact, we can obtain the following (loose) lower bound:

**Proposition 4** (lower bound on $\|w^{\text{BP}}\|_2$)**.** *When $p \leq e^{(n-1)/16}/n$, $n \geq s$, and $n \geq 17$, we have*

$$\frac{\|w^{BP}\|_2}{\|\epsilon_{train}\|_2} \geq \frac{1}{3\sqrt{2}}\sqrt{\frac{1}{\ln p}}$$

*with probability at least $1 - 3/n$.*

Although there is still a significant gap from the upper bound in Theorem 2, Proposition 4 does imply that the descent of BP cannot be faster than $1/\sqrt{\ln p}$. The proof of Proposition 4 is based on a lower-bound on $\|w^{\text{BP}}\|_1$, which is actually tight. For details, please refer to Supplementary Material (Appendix K).

**(iii) Large $n$ increases the model error of BP, but large $p$ compensates it in such a way that the descent floor is independent of $n$.** From Eq. (9), we can observe that, for a fixed $p$ inside the descent region, larger $n$ makes the performance of BP worse. This is confirmed by our numerical results in Fig. 2: when $p$ is relatively small but larger than $n$ (i.e., in the descent region), $\|w^{\text{BP}}\|_2$ is larger when $n$ increases from $n = 150$ to $n = 600$. (Note that when $p < n$, with high probability overfitting cannot happen, i.e., no solution can satisfy Eq. (7). Instead, for $p < n$ we plot the model error of the min-MSE solutions, which ascends to the peak at $p = n$ right before the descent region.) While surprising, this behavior is however reasonable. As $n$ increases, the null space corresponding to Eq. (7) becomes smaller. Therefore, more data means that BP has to "work harder" to fit the noise in those data, and consequently BP introduces larger model errors. On the positive side, while larger $n$ degrades the performance of BP, the larger $p$ (i.e., overparameterization) helps in a non-trivial way that cancels out the additional increase in model error, so that ultimately the descent floor is

independent of $n$ (see Corollary 3). This is again confirmed by Fig. 2 where the lowest points of the overfitting regime ($p > n$) of all curves for different $n$ are very close. We note that for the min $\ell_2$-norm solution, one can easily verify that its descent in Eq. (11) stops at $p = \frac{(n+1)\|\beta\|_2}{\|\beta\|_2 - \sigma}$, which leads to a descent floor for $\|\underline{w}^{\ell_2}\|_2$ at the level around $\sqrt{2\|\beta\|_2 \sigma - \sigma^2}$, independently of $n$. Further, for $p$ smaller than the above value, its model error will also increase with $n$. Thus, from this aspect both BP and min $\ell_2$-norm solutions have similar dependency on $n$. However, as we comment below, the descent floor of BP can be much wider and deeper.

**(iv) For high** SNR **and small** $s$**, the descent floor of BP is lower and wider.** Comparing the above-mentioned descent floors between BP and min $\ell_2$-norm solutions, we can see, when SNR $= \|\beta\|_2^2/\sigma^2 \gg 1$, the descent floor of BP is about $\Theta(\sqrt{\frac{1}{s}\sqrt{\mathrm{SNR}}})$ lower than that of min $\ell_2$-norm solutions. Further, since the model error of BP decreases in $\ln p$, we expect the descent floor of BP to be significantly wider than that of min $\ell_2$-norm solutions. Both are confirmed numerically in Fig. 1, suggesting that BP may produce more accurate solutions more easily for higher-SNR training data and sparser models.

Finally, Fig. 3 verifies that the quantitative dependency of the model error on $n$, $s$ and $\|\epsilon_{\text{train}}\|_2$ predicted by our upper bound in Theorem 2 is actually quite tight. For large $n$, we can see that all curves of $n^{-1/4}\|w^{\text{BP}}\|_2$ approach horizontal lines, suggesting that $\|w^{\text{BP}}\|_2$ is indeed proportional to $n^{1/4}$ in the descent region. (The deviation at small $n$ is because, for such a small $n$, $p = 5000$ has already passed the descent region.) Further, we observe that the horizontal part of the blue curve "$s = 1, \|\epsilon_{\text{train}}\|_2 = 0.15$" almost overlaps with that of the orange curve "$s = 20, \|\epsilon_{\text{train}}\|_2 = 0.15$", which matches with Theorem 2 that the descent is independent of $s$ in the descent region. Finally, the horizontal part of the green curve "$s = 20, \|\epsilon_{\text{train}}\|_2 = 0.6$" are almost 4 times as high as that of the orange curve "$s = 20, \|\epsilon_{\text{train}}\|_2 = 0.15$", which matches with Theorem 2 that the descent is proportional to $\|\epsilon_{\text{train}}\|_2$.

We note that with a careful choice of regularization parameters, LASSO can potentially drive the $\ell_2$-norm of the model error to be as low as $\Theta(\sigma\sqrt{s \log p/n})$ [27]. As we have shown above, neither BP nor min $\ell_2$-norm solutions can push the model error to be this low. However, LASSO by itself does not produce overfitting solutions, and the type of explicit regularization in LASSO (to avoid overfitting) is also not used in DNN. Thus, it remains a puzzle whether and how one can design practical overfitting solution that can reach this level of accuracy without any explicit regularization.

## 4 Main ideas of the proof

In this section, we present the main ideas behind the proof of Theorem 2, which also reveal additional insights for BP. We start with the following definition. Let $w^I$ be the solution to the following problem (recall that $w_0$ denotes the sub-vector that consists of the first $s$ elements of $w$, i.e., corresponding to the non-zero elements of the true regressor $\underline{\beta}$):

$$\min_w \|w\|_1, \quad \text{subject to } \mathbf{X}_{\text{train}}w = \epsilon_{\text{train}}, \ w_0 = \mathbf{0}. \tag{12}$$

In other words, $w^I$ is the regressor that fits only the noise $\epsilon_{\text{train}}$. Assuming that the matrix $[\mathbf{X}_{s+1} \ \mathbf{X}_{s+2} \ \cdots \ \mathbf{X}_p] \in \mathbb{R}^{n \times (p-s)}$ has full row-rank (which occurs almost surely), $w^I$ exists if and only if $p - s \geq n$, which is slightly stricter than $p > n$. The rest of this paper is based on the condition that $w^I$ exists, i.e., $p - s \geq n$. Notice that in Theorem 2, the condition $p \geq (16n)^4$ already implies that $p \geq (16n)^4 \geq 2n \geq s + n$.

The first step (Proposition 5 below) is to relate the magnitude of $w^{\text{BP}}$ with the magnitude of $w^I$. The reason that we are interested in this relationship is as follows. Note that one potential way for an overfitting solution to have a small model error is that the solution uses the $(p-s)$ "redundant" elements of the regressor to fit the noise, without distorting the $s$ "significant" elements (that correspond to the non-zero basis of the true regressor). In that case, as $(p-s)$ increases, it will be increasingly easier for the "redundant" elements of the regressor to fit the noise, and thus the model error may improve with respect to $p$. In other words, we expect that $\|w^I\|_1$ will decrease as $p$ increases. However, it is not always true that, as the "redundant" elements of the regressor fit the noise better, they do not distort the "significant" elements of the regressor. Indeed, $\ell_2$-minimization would be such a counter-example: as $p$ increases, although it is also increasingly easier for the regressor to fit the noise

[29], the "significant" elements of the regressor also go to zero [7]. This is precisely the reason why the model error of the min $\ell_2$-norm overfitting solution in Eq. (11) quickly approaches the null risk $\|\beta\|_2$ as $p$ increases. In contrast, Proposition 5 below shows that this type of undesirable distortion will not occur for BP under suitable conditions.

Specifically, define the *incoherence* of $\mathbf{X}_{\text{train}}$ [15, 16] as

$$M := \max_{i \neq j} \left| \mathbf{X}_i^T \mathbf{X}_j \right|, \tag{13}$$

where $\mathbf{X}_i$ and $\mathbf{X}_j$ denote $i$-th and $j$-th columns of $\mathbf{X}_{\text{train}}$, respectively. Thus, $M$ represents the largest absolute value of correlation (i.e., inner-product) between any two columns of $\mathbf{X}_{\text{train}}$ (recall that the $\ell_2$-norm of each column is exactly 1). Further, let

$$K := \frac{1 + M}{sM} - 4. \tag{14}$$

We then have the following proposition that relates the model error $w^{\text{BP}}$ to the magnitude of $w^I$.

**Proposition 5.** *When $K > 0$, we have*

$$\|w^{BP}\|_1 \leq \left(1 + \frac{8}{K} + 2\sqrt{\frac{1}{K}}\right) \|w^I\|_1 + \frac{2\|\epsilon_{train}\|_2}{\sqrt{KM}}. \tag{15}$$

Please refer to Supplementary Material (Appendix D) for the proof. Proposition 5 shows that, as long as $\|w^I\|_1$ is small, $\|w^{\text{BP}}\|_1$ will also be small. Note that in Eq. (13), $M$ indicates how similar any two features (corresponding to two columns of $\mathbf{X}_{\text{train}}$) are. As long as $M$ is much smaller than $1/s$, in particular if $M \leq 1/(8s)$, then the value of $K$ defined in Eq. (14) will be no smaller than 4. Then, the first term of Eq. (15) will be at most a constant multiple of $\|w^I\|_1$. In conclusion, $\|w^{\text{BP}}\|_1$ will not be much larger than $\|w^I\|_1$ as long as the columns of $\mathbf{X}_{\text{train}}$ are not very similar.

Proposition 5 only captures the $\ell_1$-norm of $w^{\text{BP}}$. Instead, the test error in Eq. (2) is directly related to the $\ell_2$-norm of $w^{\text{BP}}$. Proposition 6 below relates $\|w^{\text{BP}}\|_2$ to $\|w^{\text{BP}}\|_1$.

**Proposition 6.** *The following holds:*

$$\|w^{BP}\|_2 \leq \|\epsilon_{train}\|_2 + \sqrt{M}\|w^{BP}\|_1.$$

The proof is available in Supplementary Material (Appendix E). Note that for an arbitrary vector $\alpha \in \mathbb{R}^p$, we can only infer $\|\alpha\|_2 \leq \|\alpha\|_1$. In contrast, Proposition 6 provides a much tighter bound for $\|w^{\text{BP}}\|_2$ when $M$ is small (i.e., the similarity between the columns of $\mathbf{X}_{\text{train}}$ is low).

Combining Propositions 5 and 6, we have the following corollary that relates $\|w^{\text{BP}}\|_2$ to $\|w^I\|_1$.

**Corollary 7.** *When $K > 0$, we must have*

$$\|w^{BP}\|_2 \leq \left(1 + \frac{2}{\sqrt{K}}\right)\|\epsilon_{train}\|_2 + \sqrt{M}\left(1 + \frac{8}{K} + \frac{2}{\sqrt{K}}\right)\|w^I\|_1.$$

It remains to bound $\|w^I\|_1$ and $M$. The following proposition gives an upper bound on $\|w^I\|_1$.

**Proposition 8.** *When $n \geq 100$ and $p \geq (16n)^4$, with probability at least $1 - 2e^{-n/4}$ we have*

$$\frac{\|w^I\|_1}{\|\epsilon_{train}\|_2} \leq \sqrt{1 + \frac{3n/2}{\ln p}}. \tag{16}$$

The proof of Proposition 8 is quite involved and is available in Supplementary Material (Appendix H). Proposition 8 shows that $\|w^I\|_1$ decreases in $p$ at the rate of $O(\sqrt{n/\ln p})$. This is also the reason that $n/\ln p$ shows up in the upper bound in Theorem 2. Further, $\|w^I\|_1$ is upper bounded by a value proportional to $\|\epsilon_{\text{train}}\|_2$, which, when combined with Corollary 7, implies that $\|w^{\text{BP}}\|_2$ is on the order of $\|\epsilon_{\text{train}}\|_2$. Note that the decrease of $\|w^I\|$ in $p$ trivially follows from its definition in Eq. (12) because, when $w^I$ contains more elements, the optimal $w^I$ in Eq. (12) should only have a smaller norm. In contrast, the contribution of Proposition 8 is in capturing the exact speed with which $\|w^I\|_1$ decreases with $p$, which has not been studied in the literature. When $p$ approaches $+\infty$, the

upper bound in Eq. (16) becomes 1. Intuitively, this is because with an infinite number of features, eventually there are columns of $\mathbf{X}_{\text{train}}$ that are very close to the direction of $\epsilon_{\text{train}}$. By choosing those columns, $\|w^I\|_1$ approaches $\|\epsilon_{\text{train}}\|_2$. Finally, the upper bound in Eq. (16) increases with the number of samples $n$. As we discussed earlier, this is because as $n$ increases, there are more constraints in Eq. (12) for $w^I$ to fit. Thus, the magnitude of $w^I$ increases.

Next, we present an upper bound on $M$ as follows.

**Proposition 9.** *When* $p \leq e^{n/36}$, *with probability at least* $1 - 2e^{-\ln p} - 2e^{-n/144}$ *we have*

$$M \leq 2\sqrt{7}\sqrt{\frac{\ln p}{n}}.$$

The proof is available in Supplementary Material (Appendix J). To understand the intuition behind, note that it is not hard to verify that for any $i \neq j$, the standard deviation of $\mathbf{X}_i^T \mathbf{X}_j$ is approximately equals to $1/\sqrt{n}$. Since $M$ defined in Eq. (13) denotes the maximum over $p \times (p-1)$ such pairs of columns, it will grow faster than $1/\sqrt{n}$. Proposition 9 shows that the additional multiplication factor is of the order $\sqrt{\ln p}$. As $p$ increases, eventually we can find some columns that are close to each other, which implies that $M$ is large. When some columns among the last $(p-s)$ columns of $\mathbf{X}_{\text{train}}$ are quite similar to the first $s$ columns, $M$ will be large and BP cannot distinguish the true features from spurious features. This is the main reason that the "double descent" will eventually stop when $p$ is very large, and thus Theorem 2 only holds up to a limit of $p$.

Combining Propositions 8 and 9, we can then prove Theorem 2. Please refer to Supplementary Material (Appendix F) for details.

## 5 Conclusions and future work

In this paper, we studied the generalization power of basis pursuit (BP) in the overparameterized regime when the model overfits the data. Under a sparse linear model with *i.i.d.* Gaussian features, we showed that the model error of BP exhibits "double descent" in a way quite different from min $\ell_2$-norm solutions. Specifically, the double-descent upper-bound of BP is independent of the signal strength. Further, for high SNR and sparse models, the descent-floor of BP can be much lower and wider than that of min $\ell_2$-norm solutions.

There are several interesting directions for future work. First, the gap between our upper bound (Theorem 2) and lower bound (Proposition 4) is still large, which suggests rooms to tighten these bounds. Second, these bounds work when $p$ is much larger than $n$, e.g., $p \sim \exp(n)$. It would be useful to also understand the error bound of BP when $p$ is just a little larger than $n$ (similar to [28] but for finite $p$ and $n$). Third, we only study isotropic Gaussian features in this paper. It would be important to see if our main conclusions can also be generalized to other feature models (e.g., Fourier features [30]), models with mis-specified features [7, 19, 28], or even the 2-layer neural network models of [26]. Finally, we hope that the difference between min $\ell_1$-norm solutions and min $\ell_2$-norm solutions reported here could help us understand the generalization power of overparameterized DNNs, or lead to training methods for DNNs with even better performance in such regimes.

### Acknowledgement

This work has been supported in part by an NSF sub-award via Duke University (NSF IIS-1932630), NSF grants CAREER CNS-1943226, ECCS-1818791, CCF-1758736, CNS-1758757, CNS-1717493, ONR grant N00014-17-1-2417, and a Google Faculty Research Award.

### Broader Impact

Understanding the generalization power of heavily over-parameterized networks is one of the most foundation aspects of deep learning. By understanding the double descent of generalization errors for Basis Pursuit (BP) when overfitting occurs, our work advances the understanding of how superior generalization power can arise for overfitting solutions. Such an understanding will contribute to laying a solid theoretical foundation of deep learning, which in turn may lead to practical guidelines in applying deep learning in diverse fields such as image processing and natural languages processing.

Controlling the $\ell_1$-norm also plays a fundamental role in optimization with sparse models, which have found important applications in, e.g., compressive sensing and matrix completion. Without thoroughly studying the overparameterized regime of BP (which minimizes $\ell_1$-norm while overfitting the data), the theory of double descent remains incomplete.

The insights revealed via our proof techniques in Section 4 (i.e., the relationship between the error of fitting data and the error of fitting only noise) not only help to analyze the double descent of BP, but may also be of value to other more general models.

Potential negative impacts: Our theoretical results in this paper are the first step towards understanding the double descent of BP, and should be used with care. In particular, there is still a significant gap between our upper and lower bounds. Thus, the significance of our theories lies more in revealing the general trend rather than precise characterization of double descent. Further, the Gaussian model may also be a limiting factor. Therefore, more efforts will be needed to sharpen the bounds and generalize the results for applications in practice.

## Footnotes

[1]Such sparsity requirements are not uncommon, e.g., Theorem 3.1 of [16] also requires the sparsity to be below some function of $M$ (incoherence of $\mathbf{X}_{\text{train}}$), which is related to $n$ according to Proposition 9 in our paper.

[2]https://github.com/functionadvanced/basis_pursuit_code

[3]The direction of $\beta$ is chosen uniformly at random in all simulations.

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
