[Supplementary Material]

# A  Proof of Lemma 10

**Lemma 10.** *If Eq.* (8) *has one or multiple solutions, there must exist one with at most $n$ non-zero elements.*

*Proof.* We prove by contradiction. Suppose on the contrary that every solution to Eq. (8) has at least $(n+1)$ non-zero elements. Let $\beta_0$ denote a solution with the smallest number of non-zero elements. Let $\mathcal{A}$ denote the set of indices of non-zero elements of $\beta_0$. Then, we have $|\mathcal{A}| \geq n+1$. Below, we will show that there must exist another solution to Eq. (8) with strictly fewer non-zero elements than $\beta_0$, which leads to a contradiction. Towards this end, note that since $\mathbf{X}_{\text{train}}$ has only $n$ rows, the subset of columns $\mathbf{X}_i$, $i \in \mathcal{A}$, must be linear dependent. Therefore, we can always find a non-empty set $\mathcal{B} \in \mathcal{A}$ and coefficients $c_i \neq 0$ for $i \in \mathcal{B}$ such that

$$\sum_{i \in \mathcal{B}} c_i \mathbf{X}_i = \mathbf{0}. \tag{17}$$

Define $\beta_\lambda \in \mathbb{R}^p$ for $\lambda \in \mathbb{R}$ such that

$$\beta_\lambda[i] = \begin{cases} \beta_0[i] + \lambda c_i, & \text{if } i \in \mathcal{B}, \\ \beta_0[i], & \text{otherwise.} \end{cases}$$

Note that this definition is consistent with the definition of $\beta_0$ when $\lambda = 0$. Thus, for any $\lambda \in \mathbb{R}$, we have

$$\begin{aligned} \mathbf{X}_{\text{train}}\beta_\lambda &= \mathbf{X}_{\text{train}}\beta_0 + \lambda \sum_{j=1}^{k} c_j \mathbf{X}_{b_j} \\ &= \mathbf{X}_{\text{train}}\beta_0 \text{ (by Eq. (17))} \\ &= \mathbf{Y}_{\text{train}} \text{ (since } \beta_0 \text{ satisfies the constraint of Eq. (8)).} \end{aligned} \tag{18}$$

In other words, any $\beta_\lambda$ also satisfies the constraint of Eq. (8). Define

$$\mathcal{L} := \left\{ i \in \mathcal{B} \;\middle|\; -\frac{\beta_0[i]}{c_i} < 0 \right\}, \quad \mathcal{U} := \left\{ i \in \mathcal{B} \;\middle|\; -\frac{\beta_0[i]}{c_i} > 0 \right\},$$

$$\text{LB} := \begin{cases} \max_{i \in \mathcal{L}} \left( -\frac{\beta_0[i]}{c_i} \right), & \text{if } \mathcal{L} \neq \varnothing, \\ 0, & \text{otherwise,} \end{cases}$$

$$\text{UB} := \begin{cases} \min_{i \in \mathcal{U}} \left( -\frac{\beta_0[i]}{c_i} \right), & \text{if } \mathcal{U} \neq \varnothing, \\ 0, & \text{otherwise.} \end{cases}$$

Base on those definitions, we immediately have the following two properties for the interval $[\text{LB}, \text{UB}]$. First, we must have $[\text{LB}, \text{UB}] \neq \varnothing$. This can be proved by contradiction. Suppose on the contrary that $[\text{LB}, \text{UB}] = \varnothing$. Because by definition $\text{LB} \leq 0$ and $\text{UB} \geq 0$, we must have $\text{LB} = \text{UB} = 0$. Because $\text{LB} = 0$, we must have $\mathcal{L} = \varnothing$. Because $\text{UB} = 0$, we must have $\mathcal{U} = \varnothing$. Thus, we have $\mathcal{B} = \mathcal{L} \cup \mathcal{U} = \varnothing$, which contradicts the fact that $\mathcal{B}$ is not empty. We can thus conclude that $[\text{LB}, \text{UB}] \neq \varnothing$. Second, for any $\lambda \in (\text{LB}, \text{UB})$, $\text{sign}(\beta_0[i] + \lambda c_i) = \text{sign}(\beta_0[i])$ for all $i \in \mathcal{B}$. This is because

$$\frac{\beta_0[i] + \lambda c_i}{\beta_0[i]} = 1 - \lambda \left( -\frac{c_i}{\beta_0[i]} \right) > \begin{cases} 1 - \text{LB} \cdot \left( -\frac{c_i}{\beta_0[i]} \right) \geq 0, & \text{if } i \in \mathcal{L}, \\ 1 - \text{UB} \cdot \left( -\frac{c_i}{\beta_0[i]} \right) \geq 0, & \text{if } i \in \mathcal{U}. \end{cases}$$

By the second property, we can show that $\|\beta_\lambda\|_1$ is a linear function with respect to $\lambda$ when $\lambda \in [\text{LB}, \text{UB}]$. Indeed, we can check that $\|\beta_\lambda\|_1$ is continuous with respect to $\lambda$ everywhere and its derivative is a constant in $\lambda \in (\text{LB}, \text{UB})$, i.e.,

$$\left. \frac{\partial \|\beta_\lambda\|_1}{\partial \lambda} \right|_{\lambda \in (\text{LB}, \text{UB})} = \sum_{i \in \mathcal{B}} c_i \cdot \text{sign}(\beta_0[i] + \lambda c_i) = \sum_{i \in \mathcal{B}} c_i \cdot \text{sign}(\beta_0[i]). \tag{19}$$

By the first property, there are only three possible cases to consider.

Case 1: LB $< 0$ and UB $> 0$. By linearity, we have

$$\min\{\|\beta_{\mathsf{LB}}\|_1, \|\beta_{\mathsf{UB}}\|_1\} \le \|\beta_0\|_1.$$

Thus, by Eq. (18), we know that either $\beta_{\mathsf{LB}}$ or $\beta_{\mathsf{UB}}$ (or both of them) is a solution of Eq. (8). By the definitions of $\beta_\lambda$, LB, and UB, we know that both $\beta_{\mathsf{LB}}$ and $\beta_{\mathsf{UB}}$ have a strictly smaller number of non-zero elements than that of $\beta_0$ when LB $\ne 0$ and UB $\ne 0$. This contradicts the assumption that $\beta_0$ has the smallest number of non-zero elements.

Case 2: LB $< 0$ and UB $= 0$. Since UB $= 0$, we have $\mathcal{U} = \varnothing$, which implies that $\beta_0[i]/c_i > 0$ for all $i \in \mathcal{B}$, i.e., $\beta_0[i]$ and $c_i$ have the same sign for all $i \in \mathcal{B}$. Thus, the value of Eq. (19) is positive, i.e., $\|\beta_\lambda\|_1$ is monotone increasing with respect to $\lambda \in [\mathsf{LB}, \mathsf{UB}]$. Thus, we have $\|\beta_{\mathsf{LB}}\|_1 \le \|\beta_0\|_1$. By Eq. (18), we know that $\beta_{\mathsf{LB}}$ is a solution of Eq. (8). By the definitions of $\beta_\lambda$ and LB, we know that $\beta_{\mathsf{LB}}$ has a strictly smaller number of non-zero elements than that of $\beta_0$ when LB $\ne 0$. This contradicts the assumption that $\beta_0$ has the smallest number of non-zero elements.

Case 3: LB $= 0$ and UB $> 0$. Similar to Case 2, we can show that $\beta_{\mathsf{UB}}$ is a solution of Eq. (8) and has a strictly smaller number of non-zero elements than that of $\beta_0$. This contradicts the assumption that $\beta_0$ has the smallest number of non-zero elements.

In conclusion, all cases lead to a contradiction. The result of this lemma thus follows. $\qquad\square$

# B   An estimate of $\|\epsilon_{\mathbf{train}}\|_2$ (close to $\sigma$ with high probability)

**Lemma 11** (stated on pp. 1325 of [22]). *Let U follow a chi-square distribution with D degrees of freedom. For any positive x, we have*

$$\Pr\left(\left\{U - D \ge 2\sqrt{Dx} + 2x\right\}\right) \le e^{-x},$$
$$\Pr\left(\left\{D - U \ge 2\sqrt{Dx}\right\}\right) \le e^{-x}.$$

Notice that $n\|\epsilon_{\text{train}}\|_2^2/\sigma^2$ follows the chi-square distribution with $n$ degrees of freedom. We thus have

$$\Pr\left(\left\{\|\epsilon_{\text{train}}\|_2^2 \le 2\sigma^2\right\}\right) = 1 - \Pr\left(\left\{\frac{n\|\epsilon_{\text{train}}\|_2^2}{\sigma^2} \ge 2n\right\}\right)$$
$$= 1 - \Pr\left(\left\{\frac{n\|\epsilon_{\text{train}}\|_2^2}{\sigma^2} - n \ge n\right\}\right).$$

Now we use the fact that

$$2\sqrt{n\frac{2-\sqrt{3}}{2}n} + 2 \cdot \frac{2-\sqrt{3}}{2}n = \sqrt{n^2(4 - 2\sqrt{3})} + (2-\sqrt{3})n$$
$$= \sqrt{n^2(\sqrt{3}-1)^2} + (2-\sqrt{3})n$$
$$= (\sqrt{3}-1)n + (2-\sqrt{3})n$$
$$= n.$$

We thus have

$$\Pr\left(\left\{\|\epsilon_{\text{train}}\|_2^2 \le 2\sigma^2\right\}\right) = 1 - \Pr\left(\left\{\frac{n\|\epsilon_{\text{train}}\|_2^2}{\sigma^2} - n \ge 2\sqrt{n\frac{2-\sqrt{3}}{2}n} + 2\cdot\frac{2-\sqrt{3}}{2}n\right\}\right)$$
$$\ge 1 - \exp\left(-\frac{2-\sqrt{3}}{2}n\right) \text{ (by Lemma 11 using } x = \frac{2-\sqrt{3}}{2}n). \quad (20)$$

We also have

$$
\begin{aligned}
\Pr\left(\left\{\|\epsilon_{\text{train}}\|_2^2 \geq \frac{\sigma^2}{2}\right\}\right) &= 1 - \Pr\left(\left\{\frac{n\|\epsilon_{\text{train}}\|_2^2}{\sigma^2} \leq \frac{n}{2}\right\}\right) \\
&= 1 - \Pr\left(\left\{n - \frac{n\|\epsilon_{\text{train}}\|_2^2}{\sigma^2} \geq \frac{n}{2}\right\}\right) \\
&= 1 - \Pr\left(\left\{n - \frac{n\|\epsilon_{\text{train}}\|_2^2}{\sigma^2} \geq 2\sqrt{n\frac{n}{16}}\right\}\right) \\
&\geq 1 - \exp\left(-\frac{n}{16}\right) \text{ (by Lemma 11 using } x = n/16).
\end{aligned}
\tag{21}
$$

In other words, when $n$ is large, $\|\epsilon_{\text{train}}\|_2^2$ should be close to $\sigma^2$. As a result, in the rest of the paper, we will use $\|\epsilon_{\text{train}}\|_2^2$ as a surrogate for the noise level.

# C   Proof of Lemma 1 (distortion of $\underline{\beta}$ due to normalization of $\mathbf{X}_{\text{train}}$ is small)

From Eq. (6), it is easy to see that the amount of distortion of $\underline{\beta}$ depends on the size of $\mathbf{H}_i$ for those $i$ such that either $\underline{\beta}[i]$ or $\hat{\underline{\beta}}^{\text{BP}}[i]$ is non-zero. More precisely, we define the sets

$$
\begin{aligned}
\mathcal{A} &:= \{i: \ \underline{\beta}[i] \neq 0\} \cup \{i: \ \hat{\underline{\beta}}^{\text{BP}}[i] \neq 0\} = \{1, 2, \cdots, s\} \cup \{i: \ \hat{\underline{\beta}}^{\text{BP}}[i] \neq 0\}, \\
\mathcal{B} &:= \mathcal{A} \setminus \{1, \cdots, s\}.
\end{aligned}
$$

Notice that because $\|\hat{\underline{\beta}}^{\text{BP}}\|_0 = \|\hat{\beta}^{\text{BP}}\|_0 \leq n$, the number of elements in $\mathcal{A}$ satisfies $|\mathcal{A}| \leq s + n$. Thus, the number of elements in $\mathcal{B}$ satisfies

$$
|\mathcal{B}| = |\mathcal{A} \setminus \{1, \cdots, s\}| = |\mathcal{A}| - s \leq s + n - s = n.
\tag{22}
$$

Then, we have

$$
\begin{aligned}
\|\underline{w}^{\text{BP}}\|_2^2 = \|\hat{\underline{\beta}}^{\text{BP}} - \underline{\beta}\|_2^2 &= \sum_{i=1}^{p} \frac{n(\hat{\beta}^{\text{BP}}[i] - \beta[i])^2}{\|\mathbf{H}_i\|_2^2} \\
&= \sum_{i \in \mathcal{A}} \frac{n(\hat{\beta}^{\text{BP}}[i] - \beta[i]^2}{\|\mathbf{H}_i\|_2^2} \\
&\leq \frac{n}{\min_{i \in \mathcal{A}} \|\mathbf{H}_i\|_2^2} \sum_{i \in \mathcal{A}} (\hat{\beta}^{\text{BP}}[i] - \beta[i])^2 \\
&= \frac{n}{\min_{i \in \mathcal{A}} \|\mathbf{H}_i\|_2^2} \|\hat{\beta}^{\text{BP}} - \beta\|_2^2 \\
&= \frac{n}{\min_{i \in \mathcal{A}} \|\mathbf{H}_i\|_2^2} \|w^{\text{BP}}\|_2^2.
\end{aligned}
\tag{23}
$$

In the same way, we can get the other side of the bound:

$$
\|\underline{w}^{\text{BP}}\|_2^2 \geq \frac{n}{\max_{i \in \mathcal{A}} \|\mathbf{H}_i\|_2^2} \|w^{\text{BP}}\|_2^2.
\tag{24}
$$

Similarly, for $\ell_1$-norm, we have

$$\|\underline{w}^{\mathrm{BP}}\|_1 = \|\hat{\underline{\beta}}^{\mathrm{BP}} - \underline{\beta}\|_1 = \sum_{i=1}^{p} \frac{\sqrt{n}\left|\hat{\beta}^{\mathrm{BP}}[i] - \beta[i]\right|}{\|\mathbf{H}_i\|_2}$$

$$= \sum_{i\in\mathcal{A}} \frac{\sqrt{n}\left|\hat{\beta}^{\mathrm{BP}}[i] - \beta[i]\right|}{\|\mathbf{H}_i\|_2}$$

$$\leq \frac{\sqrt{n}}{\min_{i\in\mathcal{A}}\|\mathbf{H}_i\|_2} \sum_{i\in\mathcal{A}} |\hat{\beta}^{\mathrm{BP}}[i] - \beta[i]|$$

$$= \frac{\sqrt{n}}{\min_{i\in\mathcal{A}}\|\mathbf{H}_i\|_2} \|\hat{\beta}^{\mathrm{BP}} - \beta\|_1$$

$$= \frac{\sqrt{n}}{\min_{i\in\mathcal{A}}\|\mathbf{H}_i\|_2} \|w^{\mathrm{BP}}\|_1, \tag{25}$$

as well as

$$\|\underline{w}^{\mathrm{BP}}\|_1 \geq \frac{\sqrt{n}}{\max_{i\in\mathcal{A}}\|\mathbf{H}_i\|_2} \|w^{\mathrm{BP}}\|_1. \tag{26}$$

It only remains to bound the minimum or maximum of $\|\mathbf{H}_i\|_2^2$ over $i \in \mathcal{A}$. Intuitively, for each $i$, since $\mathbb{E}[\|\mathbf{H}_i\|_2^2] = n$, $\|\mathbf{H}_i\|_2^2$ should be close to $n$ when $n$ is large. However, here the difficulty is that we do not know which elements $i$ belong to $\mathcal{A}$. If we were to account for all possible $i = 1, 2, \cdots, p$, when $p$ is exponentially large in $n$, our bounds for the minimum and maximum of $\|\mathbf{H}_i\|_2$ would become very loose. Fortunately, for those $i = s+1, \cdots, p$ (i.e., outside of the true basis), we can show that $\|\mathbf{H}_i\|_2^2$ is independent of $\mathcal{A}$. Using this fact, we can obtain a much tighter bound on the minimum and maximum of $\|\mathbf{H}_i\|_2^2$ on $\mathcal{A}$. Towards this end, we first show the following lemma:

**Lemma 12.** $\hat{\beta}^{BP}$ *is independent of the size* $\|\mathbf{H}_i\|_2$ *of* $\mathbf{H}_i$ *for* $i \in \{s+1, \cdots, p\}$. *In other words, scaling any* $\mathbf{H}_i$ *by a non-zero value* $\alpha_i$ *for any* $i \in \{s+1, \cdots, p\}$ *does not affect* $\hat{\beta}^{BP}$.

*Proof.* Suppose that $\mathbf{H}_i$ is scaled by any $\alpha_i \neq 0$ for any $i \in \{s+1, \cdots, p\}$. We denote the new $\mathbf{H}$ matrix by $\mathbf{H}'$, i.e., $\mathbf{H}'_i = \alpha_i \mathbf{H}_i$ for some $i \in \{s+1, \cdots, p\}$. By the normalization in Eq. (4), we know that $\mathbf{X}_{\mathrm{train}}$ does not change after this scaling. Further, because $\beta[i] = 0$ for $i \in \{s+1, \cdots, p\}$, $\mathbf{Y}_{\mathrm{train}}$ is also unchanged. Therefore, the BP solution as defined in Eq. (8) will remain the same. $\square$

Let $\mathfrak{A} \subseteq \{1, \cdots, p\}$ denote any possible realization of the set $\mathcal{A}$. By Lemma 12 and noting that all $\mathbf{H}_i$'s are *i.i.d.*, we then get that, for any $h_i \in \mathbb{R}$, $i = 1, \cdots, p$, and any fixed set $\mathcal{C} \subseteq \{s+1, \cdots, p\}$,

$$\Pr\left(\left\{\mathcal{A} = \mathfrak{A}, \|\mathbf{H}_i\|_2^2 \geq h_i, i = 1, \cdots, s\right\} \,\middle|\, \left\{\|\mathbf{H}_i\|_2^2 \geq h_i, \text{for all } i \in \mathcal{C}\right\}\right)$$
$$= \Pr\left(\left\{\mathcal{A} = \mathfrak{A}, \|\mathbf{H}_i\|_2^2 \geq h_i, i = 1, \cdots, s\right\}\right). \tag{27}$$

In other words, $\mathcal{A}$ and $\|\mathbf{H}_i\|_2^2$, $i = 1, \cdots, s$ are independent of $\|\mathbf{H}_i\|_2^2$, $i = s+1, \cdots, p$. Of course, this is equivalent to stating that $\|\mathbf{H}_i\|_2^2$, $i = s+1, \cdots, p$ are independent of $\mathcal{A}$ and $\|\mathbf{H}_i\|_2^2$, $i = 1, \cdots, s$. More precisely, for any $h_i \in \mathbb{R}$, $i = 1, \cdots, p$, and any fixed set $\mathcal{C} \subseteq \{s+1, \cdots, p\}$, we have

$$\Pr\left(\left\{\|\mathbf{H}_i\|_2^2 \geq h_i, \text{for all } i \in \mathcal{C}\right\} \,\middle|\, \left\{\mathcal{A} = \mathfrak{A}, \|\mathbf{H}_i\|_2^2 \geq h_i, i = 1, \cdots, s\right\}\right)$$
$$= \frac{\Pr\left(\left\{\mathcal{A} = \mathfrak{A}, \|\mathbf{H}_i\|_2^2 \geq h_i, i = 1, \cdots, s\right\} \,\middle|\, \left\{\|\mathbf{H}_i\|_2^2 \geq h_i, \text{for all } i \in \mathcal{C}\right\}\right)}{\Pr\left(\left\{\mathcal{A} = \mathfrak{A}, \|\mathbf{H}_i\|_2^2, i = 1, \cdots, s\right\}\right)}$$
$$\cdot \Pr\left(\left\{\|\mathbf{H}_i\|_2^2 \geq h_i, \text{for all } i \in \mathcal{C}\right\}\right) \text{ (by Bayes' Theorem)}$$
$$= \Pr\left(\left\{\|\mathbf{H}_i\|_2^2 \geq h_i, \text{for all } i \in \mathcal{C}\right\}\right) \text{ (using Eq. (27)).} \tag{28}$$

Further, because all $\mathbf{H}_i$'s are *i.i.d.*, we have

$$\Pr\left(\left\{\|\mathbf{H}_i\|_2^2 \geq h_i, \text{for all } i \in \mathcal{C}\right\}\right) = \prod_{i\in\mathcal{C}} \Pr\left(\left\{\|\mathbf{H}_i\|_2^2 \geq h_i\right\}\right) = \prod_{i\in\mathcal{C}} \Pr\left(\left\{\|\mathbf{H}_1\|_2^2 \geq h_i\right\}\right).$$

Substituting back to Eq. (28), we have

$$\Pr\left(\left\{\|\mathbf{H}_i\|_2^2 \geq h_i, \text{for all } i \in \mathcal{C}\right\} \,\Big|\, \left\{\mathcal{A} = \mathfrak{A}, \|\mathbf{H}_i\|_2^2 \geq h_i, i = 1, \cdots, s\right\}\right)$$
$$= \prod_{i \in \mathcal{C}} \Pr\left(\left\{\|\mathbf{H}_1\|_2^2 \geq h_i\right\}\right). \tag{29}$$

We are now ready to bound the probability distribution of $\min_{i \in \mathcal{A}} \|\mathbf{H}_i\|_2^2$ in Eq. (23). Because $\{1, \cdots, s\} \subseteq \mathcal{A}$, we have (recalling that $\mathcal{B} = \mathcal{A} \setminus \{1, \cdots, s\}$)

$$\Pr\left(\left\{\min_{i \in \mathcal{A}} \|\mathbf{H}_i\|_2^2 \geq \frac{n}{2}\right\}\right)$$
$$= \Pr\left(\bigcap_{i \in \mathcal{A}} \left\{\|\mathbf{H}_i\|_2^2 \geq \frac{n}{2}\right\}\right)$$
$$= \Pr\left(\left\{\|\mathbf{H}_i\|_2^2 \geq \frac{n}{2}, i = 1, \cdots, s\right\}\right) \cdot \Pr\left(\bigcap_{i \in \mathcal{B}} \left\{\|\mathbf{H}_i\|_2^2 \geq \frac{n}{2}\right\} \,\Big|\, \left\{\|\mathbf{H}_i\|_2^2 \geq \frac{n}{2}, i = 1, \cdots, s\right\}\right)$$
$$= \left(1 - \Pr\left(\left\{\|\mathbf{H}_1\|_2^2 \geq \frac{n}{2}\right\}\right)\right)^s \cdot \Pr\left(\bigcap_{i \in \mathcal{B}} \left\{\|\mathbf{H}_i\|_2^2 \geq \frac{n}{2}\right\} \,\Big|\, \left\{\|\mathbf{H}_i\|_2^2 \geq \frac{n}{2}, i = 1, \cdots, s\right\}\right) \tag{30}$$

(because all $\mathbf{H}_i$'s are *i.i.d.*).

We first study the second term of the right-hand-side of Eq. (30) by conditioning on $\mathcal{A} = \mathfrak{A}$. For any possible realization $\mathfrak{A}$ of the set $\mathcal{A}$, we have

$$\Pr\left(\bigcap_{i \in \mathcal{B}} \left\{\|\mathbf{H}_i\|_2^2 \geq \frac{n}{2}\right\} \,\Big|\, \left\{\mathcal{A} = \mathfrak{A}, \|\mathbf{H}_i\|_2^2 \geq \frac{n}{2}, i = 1, \cdots, s\right\}\right)$$
$$= \Pr\left(\bigcap_{i \in \mathfrak{A} \setminus \{1, \cdots, s\}} \left\{\|\mathbf{H}_i\|_2^2 \geq \frac{n}{2}\right\} \,\Big|\, \left\{\mathcal{A} = \mathfrak{A}, \|\mathbf{H}_i\|_2^2 \geq \frac{n}{2}, i = 1, \cdots, s\right\}\right)$$
$$= \prod_{i \in \mathfrak{A} \setminus \{1, \cdots, s\}} \Pr\left(\left\{\|\mathbf{H}_1\|_2^2 \geq \frac{n}{2}\right\}\right) \quad \text{(by letting } \mathcal{C} = \mathfrak{A} \setminus \{1, \cdots, s\} \text{ in Eq. (29))}$$
$$\geq \left(1 - \Pr\left(\left\{\|\mathbf{H}_1\|_2^2 \leq \frac{n}{2}\right\}\right)\right)^n \quad \text{(by Eq. (22))}. \tag{31}$$

Since the right-hand-side of Eq. (31) is independent of $\mathfrak{A}$, we then conclude that

$$\Pr\left(\bigcap_{i \in \mathcal{B}} \left\{\|\mathbf{H}_i\|_2^2 \geq \frac{n}{2}\right\} \,\Big|\, \left\{\|\mathbf{H}_i\|_2^2 \geq \frac{n}{2}, i = 1, \cdots, s\right\}\right) \geq \left(1 - \Pr\left(\left\{\|\mathbf{H}_1\|_2^2 \leq \frac{n}{2}\right\}\right)\right)^n.$$

Substituting back to Eq. (30), we have

$$\Pr\left(\left\{\min_{i \in \mathcal{A}} \|\mathbf{H}_i\|_2^2 \geq \frac{n}{2}\right\}\right) \geq \left(1 - \Pr\left(\left\{\|\mathbf{H}_1\|_2^2 \leq \frac{n}{2}\right\}\right)\right)^{n+s}$$
$$\geq \left(1 - \Pr\left(\left\{\|\mathbf{H}_1\|_2^2 \leq \frac{n}{2}\right\}\right)\right)^{2n} \quad \text{(assuming } s \leq n\text{)}$$
$$\geq (1 - e^{-n/16})^{2n} \tag{32}$$
$$\geq 1 - 2n \cdot e^{-n/16}$$
$$= 1 - e^{-n/16 + \ln(2n)}, \tag{33}$$

where in Eq. (32), we have used results for large deviation analysis on the probability of chi-square distribution (similar to the analysis of getting Eq. (21) in Appendix B). Using similar ideas, we can

also get

$$\Pr\left(\left\{\max_{i \in \mathcal{A}} \|\mathbf{H}_i\|_2^2 \leq 2n\right\}\right) \geq \left(1 - \Pr\left(\{\|\mathbf{H}_1\|_2^2 \geq 2n\}\right)\right)^{2n}$$

$$\geq \left(1 - \exp\left(-\frac{2 - \sqrt{3}}{2}n\right)\right)^{2n} \quad \text{(similar to Eq. (20) in Appendix B)}$$

$$\geq 1 - 2n \cdot \exp\left(-\frac{2 - \sqrt{3}}{2}n\right)$$

$$= 1 - \exp\left(-\frac{2 - \sqrt{3}}{2}n + \ln(2n)\right). \tag{34}$$

Applying Eq. (33) in Eq. (23) and applying Eq. (34) in Eq. (24), we conclude that

$$\Pr\left(\left\{\|\underline{w}^{\text{BP}}\|_2 \leq \sqrt{2}\|w^{\text{BP}}\|_2\right\}\right) = \Pr\left(\{\|\underline{w}^{\text{BP}}\|_2^2 \leq 2\|w^{\text{BP}}\|_2^2\}\right)$$

$$\geq 1 - \exp\left(-\frac{n}{16} + \ln(2n)\right),$$

$$\Pr\left(\left\{\|w^{\text{BP}}\|_2 \leq \sqrt{2}\|\underline{w}^{\text{BP}}\|_2\right\}\right) = \Pr\left(\{\|\underline{w}^{\text{BP}}\|_2^2 \leq 2\|w^{\text{BP}}\|_2^2\}\right)$$

$$\geq 1 - \exp\left(-\frac{2 - \sqrt{3}}{2}n + \ln(2n)\right).$$

Applying Eq. (33) in Eq. (25) and applying Eq. (24) in Eq. (26), we conclude that

$$\Pr\left(\left\{\|\underline{w}^{\text{BP}}\|_1 \leq \sqrt{2}\|w^{\text{BP}}\|_1\right\}\right) \geq 1 - \exp\left(-\frac{n}{16} + \ln(2n)\right),$$

$$\Pr\left(\left\{\|w^{\text{BP}}\|_1 \leq \sqrt{2}\|\underline{w}^{\text{BP}}\|_1\right\}\right) \geq 1 - \exp\left(-\frac{2 - \sqrt{3}}{2}n + \ln(2n)\right).$$

The result of Lemma 1 thus follows.

## D  Proof of Proposition 5 (relationship between $\|w^{\textbf{BP}}\|_1$ and $\|w^I\|_1$)

*Proof.* Since we focus on $w^{\text{BP}}$, we rewrite BP in the form of $w^{\text{BP}}$. Notice that

$$\|\hat{\beta}^{\text{BP}}\|_1 = \|w^{\text{BP}} + \beta\|_1 = \|w_0^{\text{BP}} + \beta_0\|_1 + \|w_1^{\text{BP}}\|_1.$$

Thus, we have

$$w^{\text{BP}} = \arg\min_w \|w_0 + \beta_0\|_1 + \|w_1\|_1$$

$$\text{subject to } \mathbf{X}_{\text{train}}w = \epsilon_{\text{train}}. \tag{35}$$

Define $\mathbf{G} := \mathbf{X}_{\text{train}}{}^T \mathbf{X}_{\text{train}}$ and let $\mathbf{I}$ be the $p \times p$ identity matrix. Let $|\cdot|$ denote the operation that takes the component-wise absolute value of every element of a matrix. We have

$$\|\epsilon_{\text{train}}\|_2^2 = \|\mathbf{X}_{\text{train}}w^{\text{BP}}\|_2^2$$

$$= (w^{\text{BP}})^T \mathbf{G} w^{\text{BP}}$$

$$= \|w^{\text{BP}}\|_2^2 + (w^{\text{BP}})^T(\mathbf{G} - \mathbf{I})w^{\text{BP}}$$

$$\geq \|w^{\text{BP}}\|_2^2 - |w^{\text{BP}}|^T|\mathbf{G} - \mathbf{I}||w^{\text{BP}}|$$

$$\overset{(a)}{\geq} \|w^{\text{BP}}\|_2^2 - M|w^{\text{BP}}|^T|\mathbb{1} - \mathbf{I}||w^{\text{BP}}|$$

$$= (1 + M)\|w^{\text{BP}}\|_2^2 - M\|w^{\text{BP}}\|_1^2, \tag{36}$$

where in step (a) $\mathbb{1}$ represents a $p \times p$ matrix with all elements equal to 1, and the step holds because $\mathbf{G}$ has diagonal elements equal to 1 and off-diagonal elements no greater than $M$ in absolute value. Because $w^I$ also satisfies the constraint of (35), by the representation of $w^{\mathrm{BP}}$ in (35), we have

$$\|w_0^{\mathrm{BP}} + \beta_0\|_1 + \|w_1^{\mathrm{BP}}\|_1 \leq \|w_0^I + \beta_0\|_1 + \|w_1^I\|_1.$$

By definition (12), we have $w_0^I = \mathbf{0}$ and $\|w_1^I\|_1 = \|w^I\|_1$. Thus, we have

$$\|w_0^{\mathrm{BP}} + \beta_0\|_1 + \|w_1^{\mathrm{BP}}\|_1 \leq \|\beta_0\|_1 + \|w^I\|_1.$$

By the triangle inequality, we have $\|\beta_0\|_1 - \|w_0^{\mathrm{BP}} + \beta_0\|_1 \leq \|w_0^{\mathrm{BP}}\|_1$. Thus, we obtain

$$\|w_1^{\mathrm{BP}}\|_1 \leq \|\beta_0\|_1 - \|w_0^{\mathrm{BP}} + \beta_0\|_1 + \|w^I\|_1$$
$$\leq \|w_0^{\mathrm{BP}}\|_1 + \|w^I\|_1. \tag{37}$$

We now use (36) and (37) to establish (15). Specifically, because $w_0^{\mathrm{BP}} \in \mathbb{R}^s$, we have

$$\|w_0^{\mathrm{BP}}\|_2^2 \geq \frac{1}{s}\|w_0^{\mathrm{BP}}\|_1^2.$$

Thus, we have

$$\|w^{\mathrm{BP}}\|_2^2 \geq \|w_0^{\mathrm{BP}}\|_2^2 \geq \frac{1}{s}\|w_0^{\mathrm{BP}}\|_1^2. \tag{38}$$

Applying Eq. (37), we have

$$\|w^{\mathrm{BP}}\|_1 = \|w_1^{\mathrm{BP}}\|_1 + \|w_0^{\mathrm{BP}}\|_1 \leq 2\|w_0^{\mathrm{BP}}\|_1 + \|w^I\|_1. \tag{39}$$

Substituting Eq. (38) and Eq. (39) in Eq. (36), we have

$$\frac{1+M}{s}\|w_0^{\mathrm{BP}}\|_1^2 - M(2\|w_0^{\mathrm{BP}}\|_1 + \|w^I\|_1)^2 \leq \|\epsilon_{\mathrm{train}}\|_2^2,$$

which can be rearranged into a quadratic inequality in $\|w_0^{\mathrm{BP}}\|_1$, i.e.,

$$\left(\frac{1+M}{s} - 4M\right)\|w_0^{\mathrm{BP}}\|_1^2 - 4M\|w^I\|_1\|w_0^{\mathrm{BP}}\|_1$$
$$- \left(M\|w^I\|_1^2 + \|\epsilon_{\mathrm{train}}\|_2^2\right) \leq 0.$$

Since $K = \frac{1+M}{sM} - 4 > 0$, we have the leading coefficient $\frac{1+M}{s} - 4M = KM > 0$. Solving this quadratic inequality for $\|w_0^{\mathrm{BP}}\|_1$, we have

$$\|w_0^{\mathrm{BP}}\|_1 \leq \frac{4M\|w^I\|_1 + \sqrt{(4M\|w^I\|_1)^2 + 4KM\left(M\|w^I\|_1^2 + \|\epsilon_{\mathrm{train}}\|_2^2\right)}}{2KM}$$
$$= \frac{2\|w^I\|_1 + \sqrt{4\|w^I\|_1^2 + K(\|w^I\|_1^2 + \frac{1}{M}\|\epsilon_{\mathrm{train}}\|_2^2)}}{K}.$$

Plugging the result into Eq. (39), we have

$$\|w^{\mathrm{BP}}\|_1 \leq \frac{4\|w_1^I\|_1 + 2\sqrt{4\|w^I\|_1^2 + K(\|w^I\|_1^2 + \frac{1}{M}\|\epsilon_{\mathrm{train}}\|_2^2)}}{K} + \|w^I\|_1.$$

This expression already provides an upper bound on $\|w^{\mathrm{BP}}\|_1$ in terms of $M$ and $\|w^I\|_1$. To obtain an even simpler equation, combining $4\|w^I\|_1/K$ with $\|w^I\|_1$, and breaking the square root apart by $\sqrt{a+b+c} \leq \sqrt{a} + \sqrt{b} + \sqrt{c}$, we have

$$\|w^{\mathrm{BP}}\|_1 \leq \frac{K+4}{K}\|w^I\|_1 + \sqrt{\left(\frac{4\|w^I\|_1}{K}\right)^2} + \sqrt{\frac{4\|w^I\|_1^2}{K}}$$
$$+ \sqrt{\frac{4\|\epsilon_{\mathrm{train}}\|_2^2}{MK}}$$
$$= \left(1 + \frac{8}{K} + 2\sqrt{\frac{1}{K}}\right)\|w^I\|_1 + \frac{2\|\epsilon_{\mathrm{train}}\|_2}{\sqrt{KM}}.$$

The result of the proposition thus follows. $\qquad\square$

# E    Proof of Proposition 6 (relationship between $\|w^{\mathbf{BP}}\|_2$ and $\|w^{\mathbf{BP}}\|_1$)

*Proof.* In the proof of Proposition 5, we have already proven Eq. (36)[4]. By Eq. (36), we have

$$
\begin{aligned}
\|w^{\mathbf{BP}}\|_2 &\leq \sqrt{\frac{\|\epsilon_{\text{train}}\|_2^2 + M\|w^{\mathbf{BP}}\|_1^2}{1 + M}} \\
&\leq \sqrt{\|\epsilon_{\text{train}}\|_2^2 + M\|w^{\mathbf{BP}}\|_1^2} \\
&\leq \|\epsilon_{\text{train}}\|_2 + \sqrt{M}\|w^{\mathbf{BP}}\|_1.
\end{aligned}
$$

$\square$

# F    Proof of Theorem 2 (upper bound of model error)

The proof consists three steps. In step 1, we verify the conditions for Proposition 8 and get the estimation on $\|w^I\|_1$ by Proposition 8. In step 2, we verify the conditions for Proposition 9 and get the estimation on $M$ by Proposition 9. In step 3, we combine results in steps 1 and 2 to prove Theorem 2.

### Step 1

We first verify that the conditions for Proposition 8 are satisfied. Towards this end, from the assumption of Theorem 2 that

$$
p \in \left[ (16n)^4, \ \exp\left( \frac{n}{1792s^2} \right) \right],
$$

we have

$$
p \geq (16n)^4, \tag{40}
$$

and

$$
p \leq \exp\left( \frac{n}{1792s^2} \right) \leq e^{n/1792} \text{ (since } s \geq 1\text{)}. \tag{41}
$$

Further, from the assumption of the theorem that $s \leq \sqrt{\frac{n}{7168\ln(16n)}}$, we have

$$
n \geq s^2 \cdot 7168 \ln(16n) \geq 7168 > 100 \text{ (since } s \geq 1 \text{ and } n \geq 1\text{)}. \tag{42}
$$

Eq. (42) and Eq. (40) imply that the condition of Proposition 8 is satisfied. We thus have, from Proposition 8, with probability at least $1 - 2e^{-n/4}$,

$$
\|w^I\|_1 \leq \sqrt{1 + \frac{3n/2}{\ln p}} \|\epsilon_{\text{train}}\|_2.
$$

From Eq. (41), we have

$$
\begin{aligned}
& p \leq e^{n/1792} \leq e^{n/2} \\
\implies & 1 \leq \frac{n/2}{\ln p}.
\end{aligned}
$$

Therefore, we have

$$
\Pr\left( \left\{ \|w^I\|_1 \leq \sqrt{\frac{2n}{\ln p}} \|\epsilon_{\text{train}}\|_2 \right\} \right) \geq 1 - 2e^{-n/4}. \tag{43}
$$

**Step 2**

Note that Eq. (41) implies that the conditions of Proposition 9 is satisfied. We thus have, from Proposition 9,

$$\Pr\left(\left\{M \le 2\sqrt{7}\sqrt{\frac{\ln p}{n}}\right\}\right) \ge 1 - 2e^{-\ln p} - 2e^{-n/144}. \tag{44}$$

**Step 3**

In this step, we will combine results in steps 1 and 2 and proof the final result of Theorem 2. Towards this end, notice that for any event $A$ and any event $B$, we have

$$\Pr(\{A\} \cap \{B\}) = \Pr(\{A\}) + \Pr(\{B\}) - \Pr(\{A\} \cup \{B\})$$
$$\ge \Pr(\{A\}) + \Pr(\{B\}) - 1.$$

Thus, by Eq. (43) and Eq. (44), we have

$$\Pr\left(\left\{\|w^I\|_1 \le \sqrt{\frac{2n}{\ln p}}\|\epsilon_{\text{train}}\|_2\right\} \cap \left\{M \le 2\sqrt{7}\sqrt{\frac{\ln p}{n}}\right\}\right) \tag{45}$$

$$\ge 1 - 2e^{-n/4} - 2e^{-\ln p} - 2e^{-n/144}$$

$$\ge 1 - 6e^{-\ln p} \text{ (since } \ln p \le n/144 \le n/4 \text{ by Eq. (41))}$$

$$= 1 - 6/p.$$

It remains to show that the event in (45) implies Eq. (9). Towards this end, note that from $M \le 2\sqrt{7}\sqrt{\frac{\ln p}{n}}$, we have

$$K = \frac{1+M}{sM} - 4 \text{ (by definition in Eq. (14))}$$

$$\ge \frac{1}{sM} - 4. \tag{46}$$

From the assumption of the theorem, we have

$$\exp\left(\frac{n}{1792s^2}\right) \ge p$$

$$\implies \frac{n}{1792s^2} \ge \ln p$$

$$\implies s \le \sqrt{\frac{n}{1792 \ln p}} = \frac{1}{16\sqrt{7}}\sqrt{\frac{n}{\ln p}}. \tag{47}$$

Applying Eq. (47) to Eq. (46), we have

$$K \ge \frac{1}{\frac{1}{16\sqrt{7}}\sqrt{\frac{n}{\ln p}} \cdot 2\sqrt{7}\sqrt{\frac{\ln p}{n}}} - 4$$

$$= 8 - 4 = 4.$$

Applying

$$M \le 2\sqrt{7}\sqrt{\frac{\ln p}{n}}, \ \|w^I\|_1 \le \sqrt{\frac{2n}{\ln p}}\|\epsilon_{\text{train}}\|_2, \text{ and } K \ge 4. \tag{48}$$

to Corollary 7, we have

$$\|w^{\text{BP}}\|_2 \le 2\|\epsilon_{\text{train}}\|_2 + \sqrt{2\sqrt{7}}\left(\frac{\ln p}{n}\right)^{1/4} \cdot 4 \cdot \sqrt{\frac{2n}{\ln p}}\|\epsilon_{\text{train}}\|_2$$

$$= \left(2 + 8\left(\frac{7n}{\ln p}\right)^{1/4}\right)\|\epsilon_{\text{train}}\|_2.$$

The result of Theorem 2 thus follows.

# G   Proof of Corollary 3 (descent floor)

*Proof.* For any $a \geq 1$, we have

$$\lfloor e^a \rfloor - e^{a/2} \geq e^a - e^{a/2} - 1 = e^{a/2}(e^{a/2} - 1) - 1$$
$$\geq \sqrt{e}(\sqrt{e} - 1) - 1 = e - \sqrt{e} - 1 \approx 0.0696.$$

It implies that $\lfloor e^a \rfloor \geq e^{a/2}$ for any $a \geq 1$. Taking logarithm at both sides, we have $\ln \lfloor e^a \rfloor \geq a/2$ for any $a \geq 1$. When $s \leq \sqrt{\frac{n}{7168 \ln(16n)}}$, we have

$$\frac{n}{1792s^2} \geq 4\ln(16n) \geq 1.$$

Thus, by the choice of $p$ in the corollary, we have

$$\ln p = \ln \left\lfloor \exp\left(\frac{n}{1792s^2}\right) \right\rfloor \geq \frac{n}{3584s^2}. \tag{49}$$

Substituting Eq. (49) into Eq. (9), we have

$$\frac{\|w^{\mathrm{BP}}\|_2}{\|\epsilon_{\mathrm{train}}\|_2} \leq 2 + 8 \left(7 \times 3584s^2\right)^{1/4}$$
$$= 2 + 32\sqrt{14}\sqrt{s}.$$

$\square$

# H   Proof of Proposition 8 (upper bound of $\|w^I\|_1$)

Recall that, by the definition of $w^I$ in Eq. (12), $w^I$ is independent of the first $s$ columns of $\mathbf{X}_{\mathrm{train}}$. For ease of exposition, let $\mathbf{A}$ denote a $n \times (p - s)$ sub-matrix of $\mathbf{X}_{\mathrm{train}}$ that consists of the last $(p - s)$ columns, i.e.,

$$\mathbf{A} := [\mathbf{X}_{s+1}\ \mathbf{X}_{s+2}\ \cdots\ \mathbf{X}_p].$$

Thus, $\|w^I\|_1$ equals to the optimal objective value of

$$\min_{\alpha \in \mathbb{R}^{p-s}} \|\alpha\|_1 \text{ subject to } \mathbf{A}\alpha = \epsilon_{\mathrm{train}}. \tag{50}$$

Let $\lambda$ be a $n \times 1$ vector that denotes the Lagrangian multiplier associated with the constraint $\mathbf{A}\alpha = \epsilon_{\mathrm{train}}$. Then, the Lagrangian of the problem (50) is

$$L(\alpha, \lambda) := \|\alpha\|_1 + \lambda^T(\mathbf{A}\alpha - \epsilon_{\mathrm{train}}).$$

Thus, the dual problem is

$$\max_{\lambda} h(\lambda), \tag{51}$$

where the dual objective function is given by

$$h(\lambda) = \inf_{\alpha} L(\alpha, \lambda).$$

Let $\mathbf{A}_i$ denote the $i$-th column of $\mathbf{A}$. It is easy to verify that

$$h(\lambda) = \inf_{\alpha} L(\alpha, \lambda)$$
$$= \begin{cases} -\infty & \text{if there exists } i \text{ such that } |\lambda^T \mathbf{A}_i| > 1, \\ -\lambda^T \epsilon_{\mathrm{train}} & \text{otherwise.} \end{cases}$$

Thus, the dual problem (51) is equivalent to

$$\max_{\lambda} \lambda^T(-\epsilon_{\mathrm{train}})$$
$$\text{subject to } -1 \leq \lambda^T \mathbf{A}_i \leq 1 \text{ for all } i \in \{1, 2, \cdots, p - s\}. \tag{52}$$

This dual formulation gives the following geometric interpretation. Consider the $\mathbb{R}^n$ space that $\lambda$ and $\mathbf{A}_i$ stay in. Since $\|\mathbf{A}_i\|_2 = 1$, the constraint $-1 \leq \lambda^T \mathbf{A}_i \leq 1$ corresponds to the region between two parallel hyperplanes that are tangent to a unit hyper-sphere at $\mathbf{A}_i$ and $-\mathbf{A}_i$, respectively. Intuitively, as $p$ goes to infinity, there will be an infinite number of such hyperplanes. Since $\mathbf{A}_i$ is uniformly random on the surface of a unit hyper-sphere, as $p$ increases, more and more such random hyperplanes "wrap" around the hyper-sphere. Eventually, the remaining feasible region becomes a unit ball. This implies that the maximum value of the problem (52) becomes $\|\epsilon_{\text{train}}\|_2$ when $p$ goes to infinity and the optimal $\lambda$ is attained when $\lambda^* = -\epsilon_{\text{train}}/\|\epsilon_{\text{train}}\|_2$. Our result in Proposition 8 is also consistent with this intuition that $\|w^I\|_1 \to \|\epsilon_{\text{train}}\|_2$ as $p \to \infty$. Of course, the challenge of Proposition 8 is to establish an upper bound of $\|w^I\|_1$ even for finite $p$, which we will study below.

Another intuition from this geometric interpretation is that, among all $\mathbf{A}_i$'s, those "close" to the direction of $\pm\epsilon_{\text{train}}$ matter most, because their corresponding hyperplanes are the ones that wrap the unit hyper-sphere around the point $\lambda^* = -\epsilon_{\text{train}}/\|\epsilon_{\text{train}}\|_2$. Next, we construct an upper bound of (52) by using $q$ such "closest" $\mathbf{A}_i$'s.

Specifically, for all $i \in \{1, 2, \cdots, p-s\}$, we define

$$\mathbf{B}_i := \begin{cases} \mathbf{A}_i & \text{if } \mathbf{A}_i^T(-\epsilon_{\text{train}}) \geq 0, \\ -\mathbf{A}_i & \text{otherwise.} \end{cases}$$

Then, we sort $\mathbf{B}_i$ according to the inner product $\mathbf{B}_i^T(-\epsilon_{\text{train}})$. Let $\mathbf{B}_{(1)}, \cdots, \mathbf{B}_{(q)}$ be the $q < p - s$ vectors with the largest inner products, i.e,

$$\mathbf{B}_{(1)}^T(-\epsilon_{\text{train}}) \geq \mathbf{B}_{(2)}^T(-\epsilon_{\text{train}}) \geq \cdots \geq \mathbf{B}_{(q)}^T(-\epsilon_{\text{train}}) \geq 0. \tag{53}$$

We then relax the dual problem (52) to

$$\max_\lambda \lambda^T(-\epsilon_{\text{train}})$$
$$\text{subject to } \lambda^T \mathbf{B}_{(i)} \leq 1 \text{ for all } i \in \{1, 2, \cdots, q\}. \tag{54}$$

Note that the constraints in (54) are a subset of those in (52). Thus, the optimal objective value of (54) is an upper bound on that of (52).

Figure 4: A 3-D geometric interpretation of Problem (54).

Fig. 4 gives an geometric interpretation of (54). In Fig. 4, the gray sphere centered at the origin $O$ denotes the unit hyper-sphere in $\mathbb{R}^n$. The top (north pole) of the sphere $O$ is denoted by the point $A$. The north direction denotes the direction of $(-\epsilon_{\text{train}})$. The vector $\overrightarrow{OC}$ denotes some $\mathbf{B}_{(i)}$, $i \in \{1, \cdots, q-1\}$. The green plane is tangent to the sphere $O$ at the point $C$. Thus, the space below the green plane denotes the feasible region defined by the constraint $\lambda^T \mathbf{B}_{(1)} \leq 1$. The point $D$ denotes the intersection of the axis $\overrightarrow{OA}$ and the green plane. Similarly, the vector $\overrightarrow{OF}$ corresponds to $\mathbf{B}_{(q)}$. Note that its corresponding hyperplane (not drawn in Fig. 4) intersects the axis $\overrightarrow{OA}$ at a higher

Figure 5: When all the points lie on some hemisphere, the objective value of Problem (56) can be infinity $\lambda$ takes the direction $\overrightarrow{OF}$.

point $E$. This suggests that, by replacing the vector $\mathbf{B}_{(i)}$ in each of the constraints of (54) by another vector that has a smaller inner-product with $(-\epsilon_{\text{train}})$, the optimal objective value of (54) will be even higher. For example, in Fig. 4, the constraint corresponding to $\overrightarrow{OC}$ is replaced by that corresponding to $\overrightarrow{OB}$. This procedure is made precise below.

For each $i \in \{1, 2, \cdots, q\}$, we define

$$
\mathbf{C}_{(i)} := \frac{\sqrt{1 - \left(\frac{\mathbf{B}_{(q)}^T(-\epsilon_{\text{train}})}{\|\epsilon_{\text{train}}\|_2}\right)^2}}{\sqrt{1 - \left(\frac{\mathbf{B}_{(i)}^T(-\epsilon_{\text{train}})}{\|\epsilon_{\text{train}}\|_2}\right)^2}} \cdot \left(\mathbf{B}_{(i)} - \frac{\mathbf{B}_{(i)}^T(-\epsilon_{\text{train}})}{\|\epsilon_{\text{train}}\|_2^2}(-\epsilon_{\text{train}})\right)
$$
$$
+ \frac{\mathbf{B}_{(q)}^T(-\epsilon_{\text{train}})}{\|\epsilon_{\text{train}}\|_2^2}(-\epsilon_{\text{train}}). \tag{55}
$$

By the definition of $\mathbf{C}_{(i)}$, it is easy to verify that $\|\mathbf{C}_{(i)}\|_2 = 1$ and $\mathbf{C}_{(i)}^T(-\epsilon_{\text{train}}) = \mathbf{B}_{(q)}^T(-\epsilon_{\text{train}}) \leq \mathbf{B}_{(i)}^T(-\epsilon_{\text{train}})$, for all $i \in \{1, \cdots, q\}$. Roughly speaking, $\mathbf{C}_{(i)}$ is the point on the unit-hyper-sphere that is along the same (vertical) longitude as $\mathbf{B}_{(i)}$, but at the same (horizontal) latitude as $\mathbf{B}_{(q)}$.

Then, we can construct another problem as follows:

$$
\max_{\lambda} \lambda^T(-\epsilon_{\text{train}}) \text{ subject to}
$$
$$
\lambda^T \mathbf{C}_{(i)} \leq 1, \text{ for all } i \in \{1, 2, \cdots, q\}. \tag{56}
$$

The following lemma shows that the solution to (56) is an upper bound on that of (54).

**Lemma 13.** *The objective value of Problem (56) must be greater than or equal to that of Problem (54).*

See Appendix I.1 for the proof. We draw the geometric interpretation of the problem (56) in Fig. 5. Vectors $\overrightarrow{OD_1}$, $\overrightarrow{OD_2}$, and $\overrightarrow{OD_3}$ represent those vectors $\mathbf{C}_{(i)}$. Since all $\mathbf{C}_{(i)}$'s have the same latitude, points $D_1$, $D_2$, and $D_3$ locate on one circle centered at point $D$ (the circle is actually a hyper-sphere in $\mathbb{R}^{n-1}$). Therefore, tangent planes on those points have the same intersection point $E$ with the axis $\overrightarrow{OD}$.

We wish to argue that the vector $\overrightarrow{OE}$ is the optimal $\lambda$ for the problem (56). However, it is not always the case. Specifically, when all those $\mathbf{C}_{(i)}$'s lie on some hemisphere in $\mathbb{R}^{n-1}$, we can find a direction $\lambda$ such that $\lambda^T(-\epsilon_{\text{train}})$ goes to infinity. For example, in Fig. 5, the direction $\overrightarrow{OF}$ corresponds to such a direction of $\lambda$ that $\lambda^T(-\epsilon_{\text{train}})$ goes to infinity. Fortunately, when $q$ is large enough, the probability that all $\mathbf{C}_{(i)}$'s lie on some hemisphere in $\mathbb{R}^{n-1}$ is very small. Towards this end, we can utilize the following result from [35].

**Lemma 14** (From [35]). *Let N points be scattered uniformly at random on the surface of a sphere in an $n$-dimensional space. Then, the probability that all the points lie on some hemisphere equals to*

$$2^{-N+1}\sum_{k=0}^{n-1}\binom{N-1}{k}.$$

Applying Lemma 14 to all $q$ points $\mathbf{C}_{(1)}, \cdots, \mathbf{C}_{(q)}$ (represented by $D_1, D_2, D_3$ in Fig. 5) on the sphere in $\mathbb{R}^{n-1}$, we can quantify the probability that the situation in Fig. 5 does not happen, in which case we can then prove that the vector $\overrightarrow{OE}$ is the optimal $\lambda$ for the problem (56). Lemma 15 below summarizes this result.

**Lemma 15.** *The problem (56) achieves the optimal objective value at*

$$\lambda_* = \frac{-\epsilon_{train}}{\mathbf{B}_{(q)}^T(-\epsilon_{train})}$$

*with the probability at least*

$$1 - 2^{-q+1}\sum_{i=0}^{n-2}\binom{q-1}{i} \geq 1 - e^{-(q/4-n)}.$$

See Appendix I.2 for the proof. Letting $q = 5n$, and combining Lemmas 13 and 15, we have the following corollary.

**Corollary 16.** *The following holds*

$$\|w^I\|_1 \leq \frac{\|\epsilon_{train}\|_2^2}{\mathbf{B}_{(5n)}^T(-\epsilon_{train})}$$

*with probability at least $1 - e^{-n/4}$.*

It only remains to bound $\mathbf{B}_{(i)}(-\epsilon_{\text{train}})$. Using the fact that each $\mathbf{B}_i$ is *i.i.d.* and uniformly distributed on the unit-hyper-hemisphere in $\mathbb{R}^n$, we have the following result.

**Lemma 17.** *When $n \geq 100$ and $p \geq (16n)^4$, the following holds*

$$\mathbf{B}_{(5n)}(-\epsilon_{train}) \geq \frac{\|\epsilon_{train}\|_2}{\sqrt{1 + \frac{3n/2}{\ln p}}}$$

*with probability at least $1 - e^{-5n/4}$.*

See Appendix I.3 for the proof. Combining Corollary 16 and Lemma 17, we then obtain Proposition 8.

# I  Proofs of supporting results in Appendix H

## I.1  Proof of Lemma 13

The proof consists of two steps. In step 1, we will define an intermediate problem (57) below, and show that problem (54) is equivalent to the problem (57). In step 2, we will show that the any feasible $\lambda$ for the problem (57) is also feasible for the problem (56). The conclusion of Lemma 13 thus follows.

For step 1, the intermediate problem is defined as follows.

$$\max_{\lambda} \lambda^T(-\epsilon_{\text{train}}) \text{ subject to}$$
$$\lambda^T(-\epsilon_{\text{train}}) \geq \mathbf{B}_{(1)}^T(-\epsilon_{\text{train}}),$$
$$\lambda^T \mathbf{B}_{(i)} \leq 1 \text{ for all } i \in \{1, 2, \cdots, q\}. \tag{57}$$

In order to show that this problem is equivalent to (54), we use the following lemma.

**Lemma 18.** *The value of the problem (54) is at least $\mathbf{B}_{(1)}^T(-\epsilon_{train})$.*

*Proof.* Because $\left| \mathbf{B}_{(1)}^T \mathbf{A}_i \right| \leq \|\mathbf{B}_{(1)}\|_2 \|\mathbf{B}_{(i)}\|_2 = 1$ for all $i \in \{1, \cdots, q\}$, $\mathbf{B}_{(1)}$ is feasible for the problem (54). The result of this lemma thus follows. $\qquad\square$

By this lemma, we can add an additional constraint $\lambda^T(-\epsilon_{\text{train}}) \geq \mathbf{B}_{(1)}^T(-\epsilon_{\text{train}})$ to the problem (54) without affecting its solution. This is exactly problem (57). Thus, the problem (54) is equivalent to the intermediate problem (57), i.e., step 1 has been proven. Then, we move on to step 2. We will first use Lemma 19 to show that if $\mathbf{C}_{(i)}$ can be written in the form of

$$\mathbf{C}_{(i)} = \frac{\mathbf{B}_i + k\epsilon_{\text{train}}}{\|\mathbf{B}_{(i)} + k\epsilon_{\text{train}}\|_2}, \tag{58}$$

for some $k > 0$ and $\mathbf{C}_{(i)}^T \epsilon_{\text{train}} \leq 0$ , then any $\lambda$ that satisfies $\lambda^T \mathbf{B}_{(i)} \leq 1$ and $\lambda^T(-\epsilon_{\text{train}}) \geq \mathbf{B}_{(1)}^T(-\epsilon_{\text{train}})$ must also satisfies $\lambda^T \mathbf{C}_{(i)} \leq 1$. After that, we use Lemma 21 to show that all $\mathbf{C}_{(i)}$'s indeed can be expressed in this form. The conclusion of step 2 then follows. Towards this end, Lemma 19 is as follows.

**Lemma 19.** *For all $i \in \{1, 2, \cdots, q\}$, for any $\lambda$ that satisfy*

$$\lambda^T \mathbf{B}_i \leq 1,$$
$$\lambda^T(-\epsilon_{train}) \geq \mathbf{B}_{(1)}^T(-\epsilon_{train}),$$

*we must have*

$$\lambda^T \frac{\mathbf{B}_i + k\epsilon_{train}}{\|\mathbf{B}_i + k\epsilon_{train}\|_2} \leq 1,$$

*for any $k \geq 0$ that satisfies $(\mathbf{B}_i + k\epsilon_{train})^T \epsilon_{train} \leq 0$.*

*Proof.* We have

$$\frac{\lambda^T \mathbf{B}_i + \lambda^T k\epsilon_{\text{train}}}{\|\mathbf{B}_i + k\epsilon_{\text{train}}\|_2} \overset{(i)}{\leq} \frac{\lambda^T \mathbf{B}_i + \mathbf{B}_i^T k\epsilon_{\text{train}}}{\|\mathbf{B}_i + k\epsilon_{\text{train}}\|_2} \overset{(ii)}{=} \frac{1 + \mathbf{B}_i^T k\epsilon_{\text{train}}}{\|\mathbf{B}_i + k\epsilon_{\text{train}}\|_2}$$
$$\overset{(iii)}{\leq} \mathbf{B}_i^T \frac{\mathbf{B}_i + k\epsilon_{\text{train}}}{\|\mathbf{B}_i + k\epsilon_{\text{train}}\|_2} \overset{(iv)}{\leq} \|\mathbf{B}_i\|_2 \frac{\|\mathbf{B}_i + k\epsilon_{\text{train}}\|_2}{\|\mathbf{B}_i + k\epsilon_{\text{train}}\|_2} \overset{(v)}{=} 1.$$

Here are reasons of each step: (i) By Eq. (53), we have $\lambda^T(-\epsilon_{\text{train}}) \geq \mathbf{B}_{(1)}^T(-\epsilon_{\text{train}}) \geq \mathbf{B}_i^T(-\epsilon_{\text{train}})$. Thus, we have $\lambda^T k\epsilon_{\text{train}} \leq \mathbf{B}_i^T k\epsilon_{\text{train}}$; (ii) $\lambda^T \mathbf{B}_i \leq 1$ by the assumption of the lemma; (iii) $\mathbf{B}_i^T \mathbf{B}_i = 1$ by definition of $\mathbf{B}_i$; (iv) Cauchy–Schwarz inequality; (v) $\|\mathbf{B}_i\|_2 = \mathbf{B}_i^T \mathbf{B}_i = 1$. $\qquad\square$

Then, it only remains to prove that all $\mathbf{C}_{(i)}$'s in Eq. (55) can be expressed in the specific form described above in Eq. (58). Towards the end, we need the following lemma, which characterizes important features of $\mathbf{C}_{(i)}$.

**Lemma 20.** *For any $i \in \{1, \cdots, q\}$, we must have $\|\mathbf{C}_{(i)}\|_2 = 1$ ,and $\mathbf{C}_{(i)}^T(-\epsilon_{train}) = \mathbf{B}_{(q)}(-\epsilon_{train})$.*

*Proof.* It is easy to verify that $\mathbf{C}_{(i)}^T(-\epsilon_{\text{train}}) = \mathbf{B}_{(q)}^T(-\epsilon_{\text{train}})$. Here we show how to prove $\|\mathbf{C}_{(i)}\|_2 = 1$. Because

$$\left(\mathbf{B}_{(i)} - \frac{\mathbf{B}_{(i)}^T(-\epsilon_{\text{train}})}{\|\epsilon_{\text{train}}\|_2^2}(-\epsilon_{\text{train}})\right)^T(-\epsilon_{\text{train}}) = 0, \tag{59}$$

we know that the first and the second term on the right hand side (RHS) of Eq. (55) are orthogonal. Thus, we have

$$\|\mathbf{C}_{(i)}\|_2^2 = \|\text{1st term on the RHS of Eq. (55)}\|_2^2 + \|\text{2nd term on the RHS of Eq. (55)}\|_2^2. \tag{60}$$

By Eq. (59), we also have

$$\left\|\frac{\mathbf{B}_{(i)}^T(-\epsilon_{\text{train}})}{\|\epsilon_{\text{train}}\|_2^2}(-\epsilon_{\text{train}})\right\|_2^2 + \left\|\mathbf{B}_{(i)} - \frac{\mathbf{B}_{(i)}^T(-\epsilon_{\text{train}})}{\|\epsilon_{\text{train}}\|_2^2}(-\epsilon_{\text{train}})\right\|_2^2 = \|\mathbf{B}_{(i)}\|_2^2 = 1.$$

Notice that

$$\left\|\frac{\mathbf{B}_{(i)}^T(-\epsilon_{\text{train}})}{\|\epsilon_{\text{train}}\|_2^2}(-\epsilon_{\text{train}})\right\|_2 = \frac{\mathbf{B}_{(i)}^T(-\epsilon_{\text{train}})}{\|\epsilon_{\text{train}}\|_2}.$$

Thus, we have

$$\left\|\mathbf{B}_{(i)} - \frac{\mathbf{B}_{(i)}^T(-\epsilon_{\text{train}})}{\|\epsilon_{\text{train}}\|_2^2}(-\epsilon_{\text{train}})\right\|_2 = \sqrt{1 - \left(\frac{\mathbf{B}_{(i)}^T(-\epsilon_{\text{train}})}{\|\epsilon_{\text{train}}\|_2}\right)^2}.$$

Thus, we have

$$\|\text{1st term on the RHS of Eq. (55)}\|_2^2 = 1 - \left(\frac{\mathbf{B}_{(q)}^T(-\epsilon_{\text{train}})}{\|\epsilon_{\text{train}}\|_2}\right)^2,$$

$$\|\text{2nd term on the RHS of Eq. (55)}\|_2^2 = \left(\frac{\mathbf{B}_{(q)}^T(-\epsilon_{\text{train}})}{\|\epsilon_{\text{train}}\|_2}\right)^2.$$

Applying those to Eq. (60), we then have $\|\mathbf{C}_{(i)}\|_2 = 1$. □

Finally, the following lemma shows that $\mathbf{C}_{(i)}$ can be written in the specific form in Eq. (58).

**Lemma 21.** *Each $\mathbf{C}_{(i)}$ defined in Eq. (55) satisfies that $\mathbf{C}_{(i)}\epsilon_{train} \leq 0$ and*

$$\mathbf{C}_{(i)} = \frac{\mathbf{B}_{(i)} + k_{(i)}\epsilon_{train}}{\|\mathbf{B}_{(i)} + k_{(i)}\epsilon_{train}\|_2}, \tag{61}$$

*where*

$$k_{(i)} = \frac{\mathbf{B}_{(i)}^T(-\epsilon_{train})}{\|\epsilon_{train}\|_2^2} - \frac{\sqrt{1 - \left(\frac{\mathbf{B}_{(i)}^T(-\epsilon_{train})}{\|\epsilon_{train}\|_2}\right)^2}}{\sqrt{1 - \left(\frac{\mathbf{B}_{(q)}^T(-\epsilon_{train})}{\|\epsilon_{train}\|_2}\right)^2}} \frac{\mathbf{B}_{(q)}^T(-\epsilon_{train})}{\|\epsilon_{train}\|_2^2} \geq 0.$$

*Proof.* Using Eq. (59) again, we decompose $\mathbf{B}_{(i)}$ into two parts: one in the direction of $(-\epsilon_{\text{train}})$, the other orthogonal to $(-\epsilon_{\text{train}})$.

$$\mathbf{B}_{(i)} = \frac{\mathbf{B}_{(i)}^T(-\epsilon_{\text{train}})}{\|\epsilon_{\text{train}}\|_2^2}(-\epsilon_{\text{train}}) + \left(\mathbf{B}_{(i)} - \frac{\mathbf{B}_{(i)}^T(-\epsilon_{\text{train}})}{\|\epsilon_{\text{train}}\|_2^2}(-\epsilon_{\text{train}})\right).$$

Thus, we have

$$\mathbf{B}_{(i)} + k_{(i)}\epsilon_{\text{train}} = \frac{\sqrt{1 - \left(\frac{\mathbf{B}_{(i)}^T(-\epsilon_{\text{train}})}{\|\epsilon_{\text{train}}\|_2}\right)^2}}{\sqrt{1 - \left(\frac{\mathbf{B}_{(q)}^T(-\epsilon_{\text{train}})}{\|\epsilon_{\text{train}}\|_2}\right)^2}} \frac{\mathbf{B}_{(q)}^T(-\epsilon_{\text{train}})}{\|\epsilon_{\text{train}}\|_2^2}(-\epsilon_{\text{train}})$$
$$+ \left(\mathbf{B}_{(i)} - \frac{\mathbf{B}_{(i)}^T(-\epsilon_{\text{train}})}{\|\epsilon_{\text{train}}\|_2^2}(-\epsilon_{\text{train}})\right).$$

We then have

$$\frac{\sqrt{1 - \left(\frac{\mathbf{B}_{(q)}^T(-\epsilon_{\text{train}})}{\|\epsilon_{\text{train}}\|_2}\right)^2}}{\sqrt{1 - \left(\frac{\mathbf{B}_{(i)}^T(-\epsilon_{\text{train}})}{\|\epsilon_{\text{train}}\|_2}\right)^2}} \cdot (\mathbf{B}_{(i)} + k_{(i)}\epsilon_{\text{train}})$$

$$= \frac{\sqrt{1 - \left(\frac{\mathbf{B}_{(q)}^T(-\epsilon_{\text{train}})}{\|\epsilon_{\text{train}}\|_2}\right)^2}}{\sqrt{1 - \left(\frac{\mathbf{B}_{(i)}^T(-\epsilon_{\text{train}})}{\|\epsilon_{\text{train}}\|_2}\right)^2}} \cdot \left(\mathbf{B}_{(i)} - \frac{\mathbf{B}_{(i)}^T(-\epsilon_{\text{train}})}{\|\epsilon_{\text{train}}\|_2^2}(-\epsilon_{\text{train}})\right)$$
$$+ \frac{\mathbf{B}_{(q)}^T(-\epsilon_{\text{train}})}{\|\epsilon_{\text{train}}\|_2^2}(-\epsilon_{\text{train}})$$
$$= \mathbf{C}_{(i)}.$$

In other words, $\mathbf{C}_{(i)}$ and $\mathbf{B}_{(i)} + k_{(i)}\epsilon_{\text{train}}$ are along the same direction. Since $\|\mathbf{C}_{(i)}\|_2 = 1$, it must then also be equal to a normalized version of $\mathbf{B}_{(i)} + k_{(i)}\epsilon_{\text{train}}$, i.e.,

$$\frac{\mathbf{B}_{(i)} + k_{(i)}\epsilon_{\text{train}}}{\|\mathbf{B}_{(i)} + k_{(i)}\epsilon_{\text{train}}\|_2} = \mathbf{C}_{(i)}.$$

This verifies (61). Note that $\mathbf{C}_{(i)}\epsilon_{\text{train}} = \mathbf{B}_{(q)}\epsilon_{\text{train}} \leq 0$ by Lemma 20. It then only remains to prove $k_{(i)} \geq 0$. Towards this end, because of Eq. (53), we have

$$\mathbf{B}_{(q)}^T(-\epsilon_{\text{train}}) \leq \mathbf{B}_{(i)}^T(-\epsilon_{\text{train}})$$

$$\implies \frac{\sqrt{1 - \left(\frac{\mathbf{B}_{(i)}^T(-\epsilon_{\text{train}})}{\|\epsilon_{\text{train}}\|_2}\right)^2}}{\sqrt{1 - \left(\frac{\mathbf{B}_{(q)}^T(-\epsilon_{\text{train}})}{\|\epsilon_{\text{train}}\|_2}\right)^2}} \leq 1.$$

Thus, we have

$$k_{(i)} \geq \frac{\mathbf{B}_{(i)}^T(-\epsilon_{\text{train}})}{\|\epsilon_{\text{train}}\|_2^2} - \frac{\mathbf{B}_{(q)}^T(-\epsilon_{\text{train}})}{\|\epsilon_{\text{train}}\|_2^2} \geq 0.$$

The result of the lemma thus follows. □

Combining Lemma 19 and Lemma 21, we have proven that if $\lambda^T(-\epsilon_{\text{train}}) \geq \mathbf{B}_{(1)}^T$ and $\lambda^T\mathbf{B}_{(i)} \leq 1$, then $\lambda^T\mathbf{C}_{(i)} \leq 1$. Therefore, we have shown step 2, i.e., any feasible $\lambda$ for the problem (57) is also feasible for the problem (56). The conclusion of Lemma 13 thus follows.

## I.2 Proof of Lemma 15

First, we show that $\lambda_*$ defined in the lemma is feasible for the problem (56). Towards this end, note that because $\mathbf{C}_{(i)}^T(-\epsilon_{\text{train}}) = \mathbf{B}_{(q)}^T(-\epsilon_{\text{train}})$ (see Lemma 20) for all $i \in \{1, 2, \cdots, q\}$, we have $\lambda_*^T \mathbf{C}_{(i)} = 1$, which implies that $\lambda_*$ is feasible for the problem (56). Then, it remains to show that $\lambda_*$ is optimal for the problem (56) with probability at least $1 - e^{-q/4-n}$.

Next, we will define an event $\mathscr{A}$ with probability no smaller than

$$1 - 2^{-q+1} \sum_{i=0}^{n-2} \binom{q-1}{i}, \tag{62}$$

such that $\lambda^*$ is optimal whenever event $\mathscr{A}$ occurs. Towards this end, consider the null space of $-\epsilon_{\text{train}}$, which is defined as

$$\ker(-\epsilon_{\text{train}}) := \{\lambda \mid \lambda^T(-\epsilon_{\text{train}}) = 0\}.$$

We then decompose all $\mathbf{C}_{(i)}$'s into two components, one is in the direction of $-\epsilon_{\text{train}}$, the other is in the null space of $-\epsilon_{\text{train}}$. Specifically, we have

$$\mathbf{C}_{(i)} = \left(\mathbf{C}_{(i)} - \frac{\mathbf{C}_{(i)}^T(-\epsilon_{\text{train}})}{\|\epsilon_{\text{train}}\|_2^2}(-\epsilon_{\text{train}})\right) + \frac{\mathbf{C}_{(i)}^T(-\epsilon_{\text{train}})}{\|\epsilon_{\text{train}}\|_2^2}(-\epsilon_{\text{train}})$$

$$= \left(\mathbf{C}_{(i)} - \frac{\mathbf{C}_{(q)}^T(-\epsilon_{\text{train}})}{\|\epsilon_{\text{train}}\|_2^2}(-\epsilon_{\text{train}})\right) + \frac{\mathbf{C}_{(q)}^T(-\epsilon_{\text{train}})}{\|\epsilon_{\text{train}}\|_2^2}(-\epsilon_{\text{train}}), \tag{63}$$

where in the last step we have used $\mathbf{C}_{(i)}^T(-\epsilon_{\text{train}}) = \mathbf{C}_{(q)}^T(-\epsilon_{\text{train}})$. For conciseness, we define

$$\mathbf{D}_{(i)} := \mathbf{C}_{(i)} - \frac{\mathbf{C}_{(q)}^T(-\epsilon_{\text{train}})}{\|\epsilon_{\text{train}}\|_2^2}(-\epsilon_{\text{train}}).$$

Since $\|\mathbf{C}_{(i)}\|_2 = 1$ and $\mathbf{C}_{(i)}$ is orthogonal to $\mathbf{C}_{(i)} - \mathbf{D}_{(i)}$, we have

$$\|\mathbf{D}_{(i)}\|_2 = \sqrt{\|\mathbf{C}_{(i)}\|_2^2 - \|\mathbf{C}_{(i)} - \mathbf{D}_{(i)}\|_2^2} = \sqrt{1 - \left(\mathbf{C}_{(q)}^T(-\epsilon_{\text{train}})\right)^2}.$$

Thus, $\mathbf{D}_{(i)}$ has the same $\ell_2$-norm for all $i \in \{1, \cdots, q\}$. Therefore, $\mathbf{D}_{(1)}, \mathbf{D}_{(2)}, \cdots, \mathbf{D}_{(q)}$ can be viewed as $q$ points in a sphere in the space $\ker(-\epsilon_{\text{train}})$, which has $(n-1)$ dimensions. By Lemma 21, we know that the projections of $\mathbf{C}_{(i)}$ and $\mathbf{B}_{(i)}$ to the space $\ker(-\epsilon_{\text{train}})$ have the same direction. Because $\mathbf{B}_{(i)}$'s are uniformly distributed on the hemisphere in $\mathbb{R}^n$, their projections to $\ker(-\epsilon_{\text{train}})$ are also uniformly distributed. Therefore, $\mathbf{D}_{(i)}$'s are uniformly distributed on a $(n-1)$-dim sphere. By Lemma 14, with probability (62), there exists at least one of the vectors $\mathbf{D}_{(1)}, \mathbf{D}_{(2)}, \cdots, \mathbf{D}_{(q)}$ in any hemisphere. Let $\mathscr{A}$ denote this event with probability (62). Note that if we use a vector $\gamma \in \ker(-\epsilon_{\text{train}})$ to represent the axis of any such hemisphere in $\mathbf{R}^{n-1}$, then whether a vector $\zeta \in \ker(-\epsilon_{\text{train}})$ is on that hemisphere is totally determined by checking whether $\gamma^T \zeta > 0$. Thus, the event $\mathscr{A}$ is equivalent to, for any $\gamma \in \ker(-\epsilon_{\text{train}})$, there exists at least one of the vectors $\mathbf{D}_{(1)}, \mathbf{D}_{(2)}, \cdots, \mathbf{D}_{(q)}$ such that its inner product with $\gamma$ is positive.

We now prove the following statement that $\lambda^*$ is optimal whenever event $\mathscr{A}$ occurs. We prove by contradiction. Assume that event $\mathscr{A}$ occurs, suppose on the contrary that the maximum point is achieved at $\lambda = \mu \neq \lambda_*$ such that $\mu^T(-\epsilon_{\text{train}}) > (\lambda^*)^T(-\epsilon_{\text{train}})$. Since $\mu$ meets all constraints, we have

$$(\mu - \lambda_*)^T \mathbf{C}_{(i)} = \mu^T \mathbf{C}_{(i)} - 1 \leq 0 \text{ for all } i \in \{1, \cdots, q\}. \tag{64}$$

Comparing the objective values at $\mu$ and $\lambda_*$, we have

$$(\mu - \lambda_*)^T(-\epsilon_{\text{train}}) > 0. \tag{65}$$

Similar to the decomposition of $\mathbf{C}_{(i)}$ in Eq. (63), we decompose $(\mu - \lambda_*)$ into two components: one in the direction of $-\epsilon_{\text{train}}$ and the other in the null space of $-\epsilon_{\text{train}}$. Specifically, we have

$$(\mu - \lambda_*) = \left((\mu - \lambda_*) - \frac{(\mu - \lambda_*)^T(-\epsilon_{\text{train}})}{\|\epsilon_{\text{train}}\|_2^2}(-\epsilon_{\text{train}})\right)$$

$$+ \frac{(\mu - \lambda_*)^T(-\epsilon_{\text{train}})}{\|\epsilon_{\text{train}}\|_2^2}(-\epsilon_{\text{train}}).$$

Thus, we have

$$(\mu - \lambda_*)^T \mathbf{C}_{(i)}$$

$$= \left( (\mu - \lambda_*) - \frac{(\mu - \lambda_*)^T (-\epsilon_{\text{train}})}{\|\epsilon_{\text{train}}\|_2^2} (-\epsilon_{\text{train}}) \right)^T$$

$$\cdot \left( \mathbf{C}_{(i)} - \frac{\mathbf{C}_{(q)}^T (-\epsilon_{\text{train}})}{\|\epsilon_{\text{train}}\|_2^2} (-\epsilon_{\text{train}}) \right)$$

$$+ \frac{1}{\|\epsilon_{\text{train}}\|_2^2} \left( (\mu - \lambda_*)^T (-\epsilon_{\text{train}}) \right) \left( \mathbf{C}_{(q)}^T (-\epsilon_{\text{train}}) \right).$$

For conciseness, we define

$$\delta := (\mu - \lambda_*) - \frac{(\mu - \lambda_*)^T (-\epsilon_{\text{train}})}{\|\epsilon_{\text{train}}\|_2^2} (-\epsilon_{\text{train}}).$$

We then have

$$(\mu - \lambda_*)^T \mathbf{C}_{(i)} = \delta^T \mathbf{D}_{(i)} + \frac{1}{\|\epsilon_{\text{train}}\|_2^2} \left( (\mu - \lambda_*)^T (-\epsilon_{\text{train}}) \right) \left( \mathbf{C}_{(q)}^T (-\epsilon_{\text{train}}) \right) \geq \delta^T \mathbf{D}_{(i)}, \quad (66)$$

where the last inequality holds because $(\mu - \lambda_*)^T (-\epsilon_{\text{train}}) > 0$ (by Eq. (65)) and $\mathbf{C}_{(q)}^T (-\epsilon_{\text{train}}) = \mathbf{B}_{(q)}^T (-\epsilon_{\text{train}}) \geq 0$ (by Lemma 20 and Eq. (53)). Since $\delta \in \ker(-\epsilon_{\text{train}})$ and event $\mathscr{A}$ occurs, we can therefore find a $\mathbf{D}_{(k)}$ such that $\delta^T \mathbf{D}_{(k)} > 0$. Letting $i = k$ in Eq. (66), we then have

$$(\mu - \lambda_*)^T \mathbf{C}_{(k)} \geq \delta^T \mathbf{D}_{(k)} > 0,$$

which contradicts Eq. (64). Therefore, $\lambda^*$ must be optimal whenever event $\mathscr{A}$ occurs.

It only remains to show that the probability of event $\mathscr{A}$ given in Eq. (62) is at least $1 - e^{-(q/4-n)}$, which is proven in the following Lemma 22.

**Lemma 22.**

$$1 - 2^{-q+1} \sum_{i=0}^{n-2} \binom{q-1}{i} \geq 1 - e^{-(q/4-n)}.$$

The proof of Lemma 22 uses the following Chernoff bound.

**Lemma 23** (Chernoff bound for binomial distribution, Theorem 4(ii) in [18]). *Let $X$ be a random variable that follows the binomial distribution $B(m, \overline{p})$, where $m$ denotes the number of experiments and $\overline{p}$ denotes the probability of success for each experiment. Then*

$$\Pr \left( \{ X \leq (1-\delta) m \overline{p} \} \right) \leq \exp \left( -\frac{\delta^2 m \overline{p}}{2} \right) \text{ for all } \delta \in (0, 1).$$

**Proof of Lemma 22:** Consider a random variable $X$ with binomial distribution $B(q-1, 1/2)$. We have

$$\Pr \left( \{ X \leq n - 2 \} \right) = 2^{-q+1} \sum_{i=0}^{n-2} \binom{q-1}{i}.$$

Let

$$\delta = 1 - \frac{2(n-2)}{q-1}, \quad \text{i.e.,} \quad 1 - \delta = \frac{2(n-2)}{q-1}.$$

Applying Chernoff bound stated in the Lemma 23, we have

$$\Pr \left( \{ X \leq n - 2 \} \right) = \Pr \left( \left\{ X \leq (1-\delta) \frac{q-1}{2} \right\} \right)$$

$$\leq e^{-\delta^2 (q-1)/4}.$$

Also, we have

$$\begin{aligned}
\delta^2(q-1)/4 &= \frac{1}{4}\left(1 - \frac{2(n-2)}{q-1}\right)^2 (q-1) \\
&\geq \frac{1}{4}\left(1 - \frac{4(n-2)}{q-1}\right)(q-1) \\
&= \frac{1}{4}(q - 1 - 4(n-2)) \\
&\geq \frac{q}{4} - n.
\end{aligned}$$

Thus, we have

$$\begin{aligned}
1 - 2^{-q+1}\sum_{i=0}^{n-2}\binom{q-1}{i} &= 1 - \Pr\left(\{x \leq n-2\}\right) \\
&\geq 1 - e^{-\delta^2(q-1)/4} \\
&\geq 1 - e^{-(q/4-n)}.
\end{aligned}$$

$\blacksquare$

## I.3  Proof of Lemma 17

The proof consists of three steps. Recall that $\mathbf{B}_{(5n)}^T(-\epsilon_{\text{train}})$ ranks the $5n$-th among all $\mathbf{A}_i^T(-\epsilon_{\text{train}})$'s and $\mathbf{A}_i^T \epsilon_{\text{train}}$'s. In step 1, we first estimate the probability distribution about $\mathbf{A}_i^T(-\epsilon_{\text{train}})$. In step 2, we use the result in step 1 to estimate $\mathbf{B}_{5n}^T(-\epsilon_{\text{train}})$. In step 3, we relax and simplify the result in step 2 to get the exact result of Lemma 17. Without loss of generality[5], we let $\epsilon_{\text{train}} = [-\|\epsilon_{\text{train}}\|_2 \ 0 \ \cdots \ 0]^T$. Thus, $\mathbf{A}_i^T(-\epsilon_{\text{train}}) = \|\epsilon_{\text{train}}\|_2 \mathbf{A}_{i1}$, where $\mathbf{A}_{ij}$ denotes the $j$-th element of the $i$-th column of $\mathbf{A}$.

### Step 1

Notice that $\mathbf{A}_i$ (i.e., the $i$-th column of $\mathbf{A}$) is a normalized Gaussian random vector. We use $\mathbf{A}_i'$ to denote the standard Gaussian random vector before the normalization, i.e., $\mathbf{A}_i'$ is a $n \times 1$ vector where each element follows i.i.d. standard Gaussian distribution. Thus, we have

$$|\mathbf{A}_{i1}| = \frac{|\mathbf{A}_{i1}'|}{\|\mathbf{A}_i'\|_2} = \frac{|\mathbf{A}_{i1}'|}{\sqrt{(\mathbf{A}_{i1}')^2 + \sum_{j=2}^n (\mathbf{A}_{ij}')^2}}.$$

For any $k > 1$, we then have

$$\Pr\left(\left\{\frac{1}{|\mathbf{A}_{i1}|} \leq k\right\}\right) = \Pr\left(\left\{(\mathbf{A}_{i1}')^2 \geq \frac{\sum_{j=2}^n (\mathbf{A}_{ij}')^2}{k^2 - 1}\right\}\right). \tag{67}$$

Notice that $\sum_{j=2}^n (\mathbf{A}_{ij}')^2$ follows the chi-square distribution with $(n-1)$ degrees of freedom. When $n$ is large, $\sum_{j=2}^n (\mathbf{A}_{ij}')^2$ should be around its mean value. Further, $\mathbf{A}_{i1}'$ follows standard Gaussian distribution. Next, we use results of chi-square distribution and Gaussian distribution to estimate the distribution of $\mathbf{A}_{i1}$. The following lemma is useful for approximating a Gaussian distribution.

**Lemma 24.** *When $t \geq 0$, we have*

$$\frac{\sqrt{2/\pi}\, e^{-t^2/2}}{t + \sqrt{t^2 + 4}} \leq \Phi^c(t) \leq \frac{\sqrt{2/\pi}\, e^{-t^2/2}}{t + \sqrt{t^2 + \frac{8}{\pi}}},$$

*where $\Phi^c(\cdot)$ denotes the complementary cumulative distribution function (cdf) of standard Gaussian distribution, i.e.,*

$$\Phi^c(t) = \frac{1}{\sqrt{2\pi}}\int_t^\infty e^{-u^2/2} du.$$

*Proof.* By (7.1.13) in [1], we know that

$$\frac{1}{x+\sqrt{x^2+2}} \le e^{x^2} \int_x^\infty e^{-y^2}\,dy \le \frac{1}{x+\sqrt{x^2+\frac{4}{\pi}}} \quad (x \ge 0).$$

Let $x = t/\sqrt{2}$. We have

$$\frac{1}{\frac{t}{\sqrt{2}}+\sqrt{\frac{t^2}{2}+2}} \le e^{t^2/2} \int_{\frac{t}{\sqrt{2}}}^\infty e^{-y^2}\,dy \le \frac{1}{\frac{t}{\sqrt{2}}+\sqrt{\frac{t^2}{2}+\frac{4}{\pi}}}$$

$$\implies \frac{\sqrt{2/\pi}\, e^{-t^2/2}}{t+\sqrt{t^2+4}} \le \frac{1}{\sqrt{\pi}} \int_{\frac{t}{\sqrt{2}}}^\infty e^{-y^2}\,dy \le \frac{\sqrt{2/\pi}\, e^{-t^2/2}}{t+\sqrt{t^2+\frac{8}{\pi}}}$$

$$\implies \frac{\sqrt{2/\pi}\, e^{-t^2/2}}{t+\sqrt{t^2+4}} \le \frac{1}{\sqrt{2\pi}} \int_t^\infty e^{-\frac{z^2}{2}}\,dz \le \frac{\sqrt{2/\pi}\, e^{-t^2/2}}{t+\sqrt{t^2+\frac{8}{\pi}}} \quad (\text{let } z := \sqrt{2}y)$$

$$\implies \frac{\sqrt{2/\pi}\, e^{-t^2/2}}{t+\sqrt{t^2+4}} \le \Phi^c(t) \le \frac{\sqrt{2/\pi}\, e^{-t^2/2}}{t+\sqrt{t^2+\frac{8}{\pi}}}.$$

The result of this lemma thus follows. $\qquad\square$

The following lemma gives an estimate of the probability distribution of $\mathbf{A}_{i1}$.

**Lemma 25.**

$$\Pr\left(\left\{\frac{1}{|\mathbf{A}_{i1}|} \le k\right\}\right) \ge 2\left(1-\frac{1}{\sqrt{e}}\right)\sqrt{\frac{2}{\pi}}\frac{e^{-t^2/2}}{t+\sqrt{t^2+4}}, \tag{68}$$

*where*

$$t = \sqrt{\frac{n+\sqrt{2}\sqrt{n-1}}{k^2-1}}.$$

*Proof.* For any $m > 0$, we have

$$\Pr\left(\left\{\frac{1}{|\mathbf{A}_{i1}|} \le k\right\}\right) = \Pr\left(\left\{(\mathbf{A}'_{i1})^2 \ge \frac{\sum_{j=2}^n (\mathbf{A}'_{ij})^2}{k^2-1}\right\}\right)$$

$$\ge \Pr\left(\left\{(\mathbf{A}'_{i1})^2 \ge \frac{n-1+2\sqrt{(n-1)m}+2m}{k^2-1}\right\}\right)$$

$$\cdot \Pr\left(\left\{\sum_{j=2}^n (\mathbf{A}'_{ij})^2 \le n-1+2\sqrt{(n-1)m}+2m\right\}\right) \quad (\text{since all } \mathbf{A}'_{ij}\text{'s are } i.i.d.)$$

Notice that $\sum_{j=2}^n (\mathbf{A}'_{ij})^2$ follows chi-square distribution with $(n-1)$ degrees freedom. Applying Lemma 11, we have

$$\Pr\left(\left\{\frac{1}{|\mathbf{A}_{i1}|} \le k\right\}\right)$$

$$\ge \Pr\left(\left\{(\mathbf{A}'_{i1})^2 \ge \frac{n-1+2\sqrt{(n-1)m}+2m}{k^2-1}\right\}\right) \cdot (1-e^{-m})$$

$$= 2(1-e^{-m})\Phi^c\left(\sqrt{\frac{n-1+2\sqrt{(n-1)m}+2m}{k^2-1}}\right) \tag{69}$$

(since the distribution of $\mathbf{A}_{i1}$ is symmetric with respect to 0).

We now let $m = 1/2$ in Eq. (69). Then

$$\sqrt{\frac{n - 1 + 2\sqrt{(n-1)m} + 2m}{k^2 - 1}} = \sqrt{\frac{n + \sqrt{2(n-1)}}{k^2 - 1}} = t.$$

Applying Lemma 24, the result of this lemma thus follows. $\qquad\square$

### Step 2

Next, we estimate the distribution of $\mathbf{B}_{(5n)}^T(-\epsilon_{\text{train}})$. We first introduce a lemma below, which will be used later.

**Lemma 26.** *If $t \geq 0.5$, then $t + \sqrt{t^2 + 4} < e^{t+0.5}$.*

*Proof.* Let $f(t) = e^{t+0.5} - (t + \sqrt{t^2 + 4})$. Then $f(0.5) \approx 0.157 > 0$. We only need to prove that $df/dt \geq 0$ when $t \geq 0.5$. Indeed, when $t \geq 0.5$, we have

$$\frac{df(t)}{dt} = e^{t+0.5} - 1 - \frac{t}{\sqrt{t^2 + 4}} \geq e - 1 - 1 \geq 0 \text{ (notice that } t \leq \sqrt{t^2 + 4} \text{ for any } t\text{).}$$

$\square$

Now, we estimate $\mathbf{B}_{(5n)}^T(-\epsilon_{\text{train}})$ by the following proposition.

**Proposition 27.** *Let*

$$C = \frac{1}{5}\left(1 - \frac{1}{\sqrt{e}}\right)\sqrt{\frac{2}{\pi}} \approx 0.063. \tag{70}$$

*When $p - s \geq ne^{9/8}/C$, the following holds.*

$$\frac{\|\epsilon_{\text{train}}\|_2}{\mathbf{B}_{(5n)}^T(-\epsilon_{\text{train}})} \leq \sqrt{1 + \frac{n + \sqrt{2}\sqrt{n-1}}{\left(\sqrt{2\ln\frac{C(p-s)}{n}} - 1\right)^2}}, \tag{71}$$

*with probability at least $1 - e^{-5n/4}$.*

(Notice that, by applying this proposition in Corollary 16, Eq. (71) already suggests an upper bound of $\|w^I\|_1$.)

*Proof.* For conciseness, we use $\rho(n, k)$ to denote the right-hand-side of Eq. (68), i.e.,

$$\rho(n, k) = 10C\frac{e^{-t^2/2}}{t + \sqrt{t^2 + 4}}\bigg|_{t = \sqrt{\frac{n+\sqrt{2}\sqrt{n-1}}{k^2-1}}}.$$

Let $k$ take the value of the RHS of Eq. (71). Then, we have

$$
\begin{aligned}
t &= \sqrt{\frac{n + \sqrt{2(n-1)}}{k^2 - 1}} \\
&= \sqrt{\frac{n + \sqrt{2(n-1)}}{1 + \frac{n+\sqrt{2(n-1)}}{\left(2\sqrt{\ln\frac{C(p-s)}{n}} - 1\right)^2} - 1}} \\
&= \sqrt{2\ln\frac{C(p-s)}{n} - 1}.
\end{aligned}
\tag{72}
$$

Because $p - s \geq ne^{9/8}/C$, we have $t \geq 0.5$. By Lemma 26, we have $t + \sqrt{t^2 + 4} < e^{t+0.5}$. Thus, we have

$$
\begin{aligned}
\rho(n, k) &\geq 10C \exp\left(-\frac{t^2}{2} - t - 0.5\right) \\
&= 10C \exp\left(-\frac{1}{2}(t+1)^2\right) \\
&= 10C \frac{n}{C(p-s)} \quad \text{(using Eq. (72))} \\
&= \frac{10n}{p - s}.
\end{aligned}
\tag{73}
$$

By the definition of $\mathbf{B}_{(5n)}$ and Eq. (53), we have

$$
\Pr\left(\{\text{Eq. (71)}\}\right) = \Pr\left(\left\{\#\{i \mid i \in \{1, 2, \cdots, p - s\}, \frac{1}{|\mathbf{A}_{i1}|} \leq k\} \geq 5n\right\}\right).
\tag{74}
$$

Consider a random variable $x$ following the binomial distribution $\mathcal{B}(p - s, \rho(n, k))$. Since $\mathbf{A}_{i1}$'s are *i.i.d.* and $\Pr\left(\left\{\frac{1}{|\mathbf{A}_{i1}|} \leq k\right\}\right) \geq \rho(n, k)$, we must have

$$
\text{Eq. (74)} \geq \Pr\left(\{x \geq 5n\}\right) = 1 - \Pr\left(\{x \leq 5n - 1\}\right) \geq 1 - \Pr\left(\{x \leq 5n\}\right).
$$

It only remains to show that $\Pr\left(\{x \leq 5n\}\right) \leq e^{-5n/4}$. Applying Lemma 23, we have

$$
\begin{aligned}
\Pr\left(\{x \leq 5n\}\right) &= \Pr\left(\{x \leq (1 - \delta)(p - s)\rho(n, k)\}\right) \\
&\leq e^{-\delta^2 (p-s)\rho(n,k)/2},
\end{aligned}
\tag{75}
$$

where

$$
\delta = 1 - \frac{5n}{(p - s)\rho(n, k)} \quad \text{(so } 5n = (1 - \delta)(p - s)\rho(n, k)\text{)}.
$$

Since $(p - s)\rho(n, k) \geq 10n$ by Eq. (73), we must have $\delta \geq 0.5$. Substituting into Eq. (75), we have $\Pr\left(\{x \leq 5n\}\right) \leq \exp(-0.5^2 \cdot (10n)/2) = e^{-5n/4}$. $\qquad\square$

### Step 3

Notice that by utilizing Proposition 27 and Corollary 16, we already have an upper bound on $\|w^I\|_1$. To get the simpler form in Lemma 17, we only need to use the following lemma to simplify the expression in Proposition 27.

**Lemma 28.** *When $n \geq 100$ and $p \geq (16n)^4$, we must have*

$$
\textit{RHS of Eq. (71)} \leq \sqrt{1 + \frac{3n/2}{\ln p}}.
$$

*Proof.* Because $n > 100$ and $p \geq (16n)^4$, we have $p \geq 10^{12}$. Thus, we ahave

$$
\begin{aligned}
&\ln p \geq 25 \text{ (since } \ln 10 \approx 2.3 > 25/12) \\
\implies &\sqrt{\ln p} - 2 \geq 3 \\
\implies &\sqrt{\ln p} - 2 \geq \sqrt{3 \ln 2 + 6} \text{ (since } \ln 2 < 1) \\
\implies &\frac{1}{2}\left(\sqrt{\ln p} - 2\right)^2 \geq \frac{3}{2}\ln 2 + 3 \\
\implies &\frac{3}{2}(\ln p - \ln 2) \geq \ln p + 2\sqrt{\ln p} + 1 \text{ (by expanding the square and rearranging terms)} \\
\implies &\sqrt{\ln p} + 1 \leq \sqrt{\frac{3}{2}}\sqrt{\ln p - \ln 2} \quad \text{(by taking square root on both sides).}
\end{aligned}
$$

Because $s \le n$ and $p \ge (16n)^4 \ge 2n$, we have $\ln(p-s) \ge \ln(p-n) \ge \ln(p/2)$. Thus, we have

$$\sqrt{\ln p} + 1 \le \sqrt{\frac{3}{2}} \sqrt{\ln(p-s)}. \tag{76}$$

We still use $C$ defined in Eq. (70). We have

$$p \ge (16n)^4 \implies p \ge \left(\frac{n}{C}\right)^4 + n + \left((16n)^4 - \left(\frac{n}{C}\right)^4 - n\right). \tag{77}$$

Note that

$$(16n)^4 - \left(\frac{n}{C}\right)^4 - n = n\left(n^3\left(16^4 - \left(\frac{1}{C}\right)^4\right) - 1\right)$$

$$\ge n\left(n^3 - 1\right) \text{ (because } 16^4 - \left(\frac{1}{C}\right)^4 \approx 16^4 - \left(\frac{1}{0.063}\right)^4 > 1)$$

$$\ge 0 \text{ (because } n \ge 1).$$

Applying it in Eq. (77), we have

$$p - n \ge \left(\frac{n}{C}\right)^4$$

$$\implies p - s \ge \left(\frac{n}{C}\right)^4 \text{ (because } s \le n)$$

$$\implies (p-s)^{-3}\left(\frac{C}{n}\right)^4 (p-s)^4 \ge 1$$

$$\implies -3\ln(p-s) + 4\ln\frac{C(p-s)}{n} \ge 0$$

$$\implies 2\ln\frac{C(p-s)}{n} \ge \frac{3}{2}\ln(p-s)$$

$$\implies 2\ln\frac{C(p-s)}{n} \ge (\sqrt{\ln p} + 1)^2 \text{ (by Eq. (76))}$$

$$\implies \left(\sqrt{2\ln\frac{C(p-s)}{n}} - 1\right)^2 \ge \ln p. \tag{78}$$

When $n \ge 100$, we always have

$$n - 1 \le \frac{n^2}{8}$$

$$\implies \sqrt{2}\sqrt{n-1} \le \frac{n}{2}. \tag{79}$$

Substituting Eq. (78) and Eq. (79) into the RHS of Eq. (71), the conclusion of this lemma thus follows. $\square$

## J  Proof of Proposition 9 (upper bound of $M$)

For conciseness, we define $G_{ij} := \mathbf{X}_i^T \mathbf{X}_j$. According to the normalization in Eq. (4), we have

$$G_{ij} := \frac{\mathbf{H}_i^T \mathbf{H}_j}{\|\mathbf{H}_i\|_2 \|\mathbf{H}_j\|_2}.$$

Our proof consists of four steps. In step 1, we relate the tail probability of any $|G_{ij}|$ (where $i \ne j$) to the tail probability of $\mathbf{H}_i^T \mathbf{H}_j$. In step 2, we estimate the tail probability of $\mathbf{H}_i^T \mathbf{H}_j$. In step 3, we use union bound to estimate the cdf of $M$, so that we can get an upper bound on $M$ with high probability. In step 4, we simplify the result derived in step 3.

**Step 1: Relating the tail probability of $|G_{ij}|$ to that of $\mathbf{H}_i^T \mathbf{H}_j$.**

For any $i \neq j$, we have

$$\Pr\left(\{|G_{ij}| > a\}\right)$$
$$= \Pr\left(\left\{|G_{ij}| > a, \|\mathbf{H}_i\|_2 \geq \sqrt{\frac{n}{2}}, \|\mathbf{H}_j\|_2 \geq \sqrt{\frac{n}{2}}\right\}\right)$$
$$+ \Pr\left(\left\{|G_{ij}| > a, \left(\|\mathbf{H}_i\|_2 < \sqrt{\frac{n}{2}} \text{ or } \|\mathbf{H}_j\|_2 < \sqrt{\frac{n}{2}}\right)\right\}\right). \tag{80}$$

The first term can be bounded by

$$\Pr\left(\left\{|G_{ij}| > a, \|\mathbf{H}_i\|_2 \geq \sqrt{\frac{n}{2}}, \|\mathbf{H}_j\|_2 \geq \sqrt{\frac{n}{2}}\right\}\right) \leq \Pr\left(\left\{|\mathbf{H}_i^T \mathbf{H}_j| > \frac{na}{2}\right\}\right),$$

because

$$|G_{ij}| > a, \|\mathbf{H}_i\|_2 \geq \sqrt{\frac{n}{2}}, \|\mathbf{H}_j\|_2 \geq \sqrt{\frac{n}{2}} \implies |\mathbf{H}_i^T \mathbf{H}_j| > \frac{na}{2}.$$

Thus, we have, from Eq. (80),

$$\Pr\left(\{|G_{ij}| > a\}\right) \leq \Pr\left(\left\{|\mathbf{H}_i^T \mathbf{H}_j| > \frac{na}{2}\right\}\right) + \Pr\left(\left\{\|\mathbf{H}_i\|_2 < \sqrt{\frac{n}{2}}\right\}\right)$$
$$+ \Pr\left(\left\{\|\mathbf{H}_j\|_2 < \sqrt{\frac{n}{2}}\right\}\right) \tag{81}$$
$$= 2\Pr\left(\left\{\mathbf{H}_i^T \mathbf{H}_j > \frac{na}{2}\right\}\right) + 2\Pr\left(\left\{\|\mathbf{H}_i\|_2 < \sqrt{\frac{n}{2}}\right\}\right),$$

where the last equality is because the distribution of $\mathbf{H}_i^T \mathbf{H}_j$ is symmetric around 0, and $\mathbf{H}_j$ has the same distribution as $\mathbf{H}_i$. Notice that $\|\mathbf{H}_i\|_2^2$ follows chi-square distribution with $n$ degrees of freedom. By Lemma 11 (using $x = n/16$), we have

$$\Pr\left(\left\{\|\mathbf{H}_i\|_2 < \sqrt{\frac{n}{2}}\right\}\right) = \Pr\left(\left\{\|\mathbf{H}_i\|_2^2 < \frac{n}{2}\right\}\right) \leq e^{-n/16}.$$

Thus, we have

$$\Pr\left(\{|G_{ij}| > a\}\right) \leq 2\Pr\left(\left\{\mathbf{H}_i^T \mathbf{H}_j > \frac{na}{2}\right\}\right) + 2e^{-n/16}. \tag{82}$$

**Step 2: Estimating the tail probability of $\mathbf{H}_i^T \mathbf{H}_j$.**

Notice that $\mathbf{H}_i^T \mathbf{H}_j$ is the sum of product of two Gaussian random variables. We will use the Chernoff bound to estimate its tail probability. Towards this end, we first calculate the moment generating function (M.G.F) of the product of two Gaussian random variables.

**Lemma 29.** *If $X$ and $Y$ are two independent standard Gaussian random variables, then the M.G.F of $XY$ is*

$$\mathbb{E}[e^{tXY}] = \frac{1}{\sqrt{1 - t^2}},$$

*for any $t^2 < 1$.*

*Proof.*

$$\mathbb{E}[e^{tXY}]$$

$$=\frac{1}{2\pi}\int_{-\infty}^{\infty}\int_{-\infty}^{\infty}e^{txy}e^{-\frac{x^2+y^2}{2}}dxdy$$

$$=\frac{1}{\sqrt{2\pi}}\int_{-\infty}^{\infty}e^{-\frac{x^2}{2}(1-t^2)}\left(\frac{1}{\sqrt{2\pi}}\int_{-\infty}^{\infty}e^{-\frac{(y-tx)^2}{2}}dy\right)dx$$

$$=\frac{1}{\sqrt{2\pi}}\int_{-\infty}^{\infty}e^{-\frac{x^2}{2}(1-t^2)}dx$$

$$=\frac{1}{\sqrt{1-t^2}}.$$

$\square$

We introduce the following lemma that helps in our calculation later.

**Lemma 30.** *For any* $x > 0$,

$$\arg\max_{t\in(0,1)}\left(tx+\frac{n}{2}\ln(1-t^2)\right)=\frac{-n+\sqrt{n^2+4x^2}}{2x}.$$

*Proof.* Let

$$f(t)=tx+\frac{n}{2}\ln(1-t^2),\quad t\in(0,1).$$

Then, we have

$$\frac{df(t)}{dt}=x-\frac{nt}{1-t^2}.$$

Letting $df(t)/dt=0$, we have exactly one solution in $(0,1)$ given by

$$t=\frac{-n+\sqrt{n^2+4x^2}}{2x}.$$

Notice that $df(t)/dt$ is monotone decreasing with respect to $t$ and thus $f(t)$ is concave on $(0,1)$. The result of this lemma thus follows. $\square$

We then use the Chernoff bound to estimate $\mathbf{H}_i^T\mathbf{H}_j$ in the following lemma.

**Lemma 31.**

$$\Pr\left(\left\{\mathbf{H}_i^T\mathbf{H}_j>\frac{na}{2}\right\}\right)$$

$$\leq\exp\left(-\frac{n}{2}\left(at+\ln\frac{2t}{a}\right)\right),$$

*where*

$$t=\frac{-1+\sqrt{1+a^2}}{a}.$$

*Proof.* Notice that

$$\mathbf{H}_i^T\mathbf{H}_j=\sum_{k=1}^{n}\mathbf{H}_{ik}\mathbf{H}_{jk}=\sum_{k=1}^{n}Z_k,$$

where $Z_k:=\mathbf{H}_{ik}\mathbf{H}_{jk}$. Using the Chernoff bound, we have

$$\Pr\left(\left\{\mathbf{H}_i^T\mathbf{H}_j>x\right\}\right)\leq\min_{t>0}e^{-tx}\prod_{k=1}^{n}\mathbb{E}[e^{tZ_k}]$$

Since each $Z_k$ is the product of two independent standard Gaussian variable, using Lemma 30, we have, for any $x > 0$,

$$\Pr\left(\{\mathbf{H}_i^T \mathbf{H}_j > x\}\right) \leq \min_{t>0} e^{-tx}(1-t^2)^{-\frac{n}{2}}$$

$$= \min_{t\in(0,1)} e^{-tx}(1-t^2)^{-\frac{n}{2}}$$

$$= \min_{t\in(0,1)} e^{-tx-\frac{n}{2}\ln(1-t^2)}$$

$$= \exp\left(-tx - \frac{n}{2}\ln(1-t^2)\right)\Big|_{t=\frac{-n+\sqrt{n^2+4x^2}}{2x}} \quad \text{(by Lemma 30)}$$

$$= \exp\left(-tx - \frac{n}{2}\ln(nt/x)\right)\Big|_{t=\frac{-n+\sqrt{n^2+4x^2}}{2x}},$$

where the last equality is because $t = (-n + \sqrt{n^2 + 4x^2})/2x$ is one solution of the quadratic equation in $t$ that $xt^2 + nt - x = 0$ (which implies $1 - t^2 = nt/x$).

Letting $x = \frac{na}{2}$, we get $t = (-1 + \sqrt{1+a^2})/a$, and

$$\exp\left(-tx - \frac{n}{2}\ln(nt/x)\right) = \exp\left(-\frac{nat}{2} - \frac{n}{2}\ln\frac{2t}{a}\right) = \exp\left(-\frac{n}{2}\left(at + \ln\frac{2t}{a}\right)\right).$$

The result of this lemma thus follows. $\qquad\square$

**Step 3: Estimating the distribution of $M$.**

Since $M$ is defined as the maximum of all $|G_{ij}|$ for $i \neq j$, we use the union bound to estimate the distribution of $M$ in the following proposition.

**Proposition 32.**

$$\Pr\left(\left\{M \leq 2\sqrt{6}\sqrt{\frac{\ln p}{n}\left(\frac{6\ln p}{n} + 1\right)}\right\}\right) \geq 1 - 2e^{-\ln p} - 2e^{-n/16 + 2\ln p}.$$

To prove Proposition 32, we introduce a technique lemma first.

**Lemma 33.** *For any $x > 0$, we must have*

$$\ln x \geq 1 - \frac{1}{x}.$$

*Proof.* We define a function

$$f(x) := \ln x - (1 - \frac{1}{x}), \quad x > 0.$$

It suffices to show that $\min f(x) = 0$. We have

$$\frac{df(x)}{dx} = \frac{1}{x} - \frac{1}{x^2} = \frac{x-1}{x^2}.$$

Thus, $f(x)$ is monotone decreasing in $(0, 1)$ and monotone increasing in $(1, \infty)$. Thus, $\min f(x) = f(1) = 0$. The conclusion of this lemma thus follows. $\qquad\square$

We are now ready to prove Proposition 32.

**Proof of Proposition 32:** Applying Lemma 31 to Eq. (82), we have

$$\Pr\left(\{|G_{ij}| > a\}\right) \leq 2\exp\left(-\frac{n}{2}\left(at + \ln\frac{2t}{a}\right)\right) + 2e^{-n/16}, \tag{83}$$

where

$$t = \frac{-1 + \sqrt{1+a^2}}{a}. \tag{84}$$

Since $M = \max_{i \neq j} |G_{ij}|$, we have

$$\Pr\left(\{M \le a\}\right)$$

$$= 1 - \Pr\left(\bigcup_{i \neq j} \{|G_{ij}| > a\}\right)$$

$$\ge 1 - \sum_{i \neq j} \Pr\left(\{|G_{ij}| > a\}\right) \text{ (by the union bound)}$$

$$= 1 - p(p-1)\Pr\left(\{|G_{ij}| > a\}\right) \text{ (since all } G_{ij} \text{ has the same distribution)}$$

$$\ge 1 - e^{2\ln p}\Pr\left(\{|G_{ij}| > a\}\right)$$

$$\ge 1 - 2e^{-n/16 + 2\ln p}$$

$$- 2\exp\left(-\frac{n}{2}\left(at + \ln\frac{2t}{a} - \frac{4\ln p}{n}\right)\right) \text{ (by Eq. (83)).} \tag{85}$$

Let

$$a = 2\sqrt{6}\sqrt{\frac{\ln p}{n}\left(\frac{6\ln p}{n} + 1\right)}. \tag{86}$$

Substituting Eq. (86) into Eq. (84), we have

$$at = -1 + \sqrt{1 + a^2}$$

$$= -1 + \sqrt{1 + \frac{24\ln p}{n} + \left(\frac{12\ln p}{n}\right)^2}$$

$$= -1 + \sqrt{\left(\frac{12\ln p}{n} + 1\right)^2}$$

$$= \frac{12\ln p}{n}. \tag{87}$$

Thus, we have

$$\ln\frac{2t}{a} = \ln\frac{2at}{a^2} = \ln\frac{2 \cdot \frac{12\ln p}{n}}{24 \cdot \frac{\ln p}{n}\left(\frac{6\ln p}{n} + 1\right)} = \ln\frac{1}{\frac{6\ln p}{n} + 1}$$

$$\ge 1 - \left(\frac{6\ln p}{n} + 1\right) \text{ (by Lemma 33)}$$

$$= -\frac{6\ln p}{n}. \tag{88}$$

By Eq. (87) and Eq. (88), we have

$$-\frac{n}{2}\left(at + \ln\frac{2t}{a} - \frac{4\ln p}{n}\right) \le -\frac{n}{2}\left(\frac{12\ln p}{n} - \frac{6\ln p}{n} - \frac{4\ln p}{n}\right) = -\ln p.$$

Substituting into Eq. (85), the result of this proposition follows. ∎

### Step 4: Simplifying the expression in Proposition 32.

By the assumption of Proposition 9 that $p \le \exp(n/36)$, we have

$$\frac{6\ln p}{n} + 1 \le \frac{7}{6}.$$

Thus, we have

$$2\sqrt{6}\sqrt{\frac{\ln p}{n}\left(\frac{6\ln p}{n} + 1\right)} \le 2\sqrt{7}\sqrt{\frac{\ln p}{n}}. \tag{89}$$

We also have

$$\frac{-n}{16} + 2\ln p \le \frac{-n}{16} + 2 \cdot \frac{n}{36} = -\frac{n}{144}. \tag{90}$$

Applying Eq. (89) and Eq. (90) to Proposition 32, we then get Proposition 9.

# K Lower bounds

In this section, we first establish a lower bound on $\|w^I\|_1$. This lower bound now only shows that our upper bound in Prop. 8 is tight (up to a constant factor), but can also be used to derive a lower bound on $\|w^{\mathrm{BP}}\|_1$. We will then use this lower bound on $\|w^{\mathrm{BP}}\|_1$ to prove Prop. 4 (i.e., the lower bound on $\|w^{\mathrm{BP}}\|_2$). As we discussed in the main body of the paper, although our bounds on $\|w^{\mathrm{BP}}\|_2$ are not tight, the bounds on $\|w^{\mathrm{BP}}\|_1$ are in fact tight (up to a constant factor), which will be shown below.

## K.1 Lower bound on $\|w^I\|_1$

A trivial lower bound on $\|w^I\|_1$ is $\|w^I\|_1 \geq \|\epsilon_{\mathrm{train}}\|_2$. To see this, letting $w^I_{(i)}$ denote the $i$-th element of $w^I$, we have

$$\|\epsilon_{\mathrm{train}}\|_2 = \|\mathbf{X}_{\mathrm{train}} w^I\|_2 = \left\|\sum_{i=1}^p w^I_{(i)} \mathbf{X}_i\right\|_2$$

$$\leq \sum_{i=1}^p |w^I_{(i)}| \cdot \|\mathbf{X}_i\|_2 = \|w^I\|_1 \text{ (notice } \|\mathbf{X}_i\|_2 = 1\text{)}.$$

Even by this trivial lower bound, we immediately know that our upper bound on $\|w^I\|_1$ in Proposition 8 is accurate when $p \to \infty$. Still, we can do better than this trivial lower bound, as shown in Proposition 35 below.

Towards this end, following the construction of Problem (54), it is not hard to show that $\mathbf{B}_{(1)}$, i.e., the vector that has the largest inner-product with $(-\epsilon_{\mathrm{train}})$, defines a lower bound for $\|w^I\|_1$.

**Lemma 34.**

$$\|w^I\|_1 \geq \frac{\|\epsilon_{train}\|_2^2}{\mathbf{B}_{(1)}^T(-\epsilon_{train})}$$

*Proof.* Let

$$\lambda_* = \frac{(-\epsilon_{\mathrm{train}})}{\mathbf{B}_{(1)}^T(-\epsilon_{\mathrm{train}})}.$$

By the definition of $\mathbf{B}_{(1)}$, for any $i \in \{1, 2, \cdots, p-s\}$, we have

$$\left|\lambda_*^T \mathbf{A}_i\right| = \frac{\left|\mathbf{A}_i^T \epsilon_{\mathrm{train}}\right|}{\left|\mathbf{B}_{(1)}^T \epsilon_{\mathrm{train}}\right|} \leq 1.$$

In other words, $\lambda_*$ satisfies all constraints of the problem (52), which implies that the optimal objective value of (52) is at least

$$\lambda_*^T(-\epsilon_{\mathrm{train}}) = \frac{\|\epsilon_{\mathrm{train}}\|_2^2}{\mathbf{B}_{(1)}^T(-\epsilon_{\mathrm{train}})}.$$

The result of this lemma thus follows. $\qquad\qquad\square$

By bounding $\mathbf{B}_{(1)}^T(-\epsilon_{\mathrm{train}})$, we can show the following result.

**Proposition 35.** *When $p \leq e^{(n-1)/16}/n$ and $n \geq 17$, then*

$$\frac{\|w^I\|_1}{\|\epsilon_{train}\|_2} \geq \sqrt{1 + \frac{n}{9 \ln p}}$$

*with probability at least $1 - 3/n$.*

The proof is available in Appendix K.4. Comparing Proposition 8 with Proposition 35, we can see that, with high probability, the upper and lower bounds of $\|w\|_1$ differ by at most a constant factor.

## K.2 Lower bounds on $\|w^{\mathbf{BP}}\|_1$ and $\|w^{\mathbf{BP}}\|_2$

Using Prop. 35, we can show the following lower bound on $\|w^{\mathrm{BP}}\|_1$.

**Proposition 36** (lower bound on $\|w^{\mathrm{BP}}\|_1$). *When $p \leq e^{(n-1)/16}/n$ and $n \geq 17$, then*

$$\|w^{BP}\|_1 \geq \frac{1}{3}\sqrt{\frac{n}{\ln p}}\|\epsilon_{train}\|_2$$

*with probability at least $1 - 3/n$.*

*Proof.* We define $w^J$ as the solution to the following optimization problem:

$$\min_w \|w\|_1, \quad \text{subject to } \mathbf{X}_{\mathrm{train}}w = \epsilon_{\mathrm{train}}.$$

By definition, $\mathbf{X}_{\mathrm{train}}w^{\mathrm{BP}} = \epsilon_{\mathrm{train}}$. Thus, we have $\|w^{\mathrm{BP}}\|_1 \geq \|w^J\|_1$. To get a lower bound on $\|w^J\|_1$, we can directly use the result in Proposition 35 because the definitions of $w^I$ and $w^J$ are essentially the same[6]. We then have,

$$\|w^J\|_1 \geq \sqrt{1 + \frac{n}{9\ln p}}\|\epsilon_{\mathrm{train}}\|_2 \geq \frac{1}{3}\sqrt{\frac{n}{\ln p}}\|\epsilon_{\mathrm{train}}\|_2$$

with probability at least $1 - 3/n$. The result of this proposition thus follows. $\square$

Next, we proceed to prove Proposition 4, i.e., the lower bound on $\|w^{\mathrm{BP}}\|_2$. Because $\|w^{\mathrm{BP}}\|_0 = \|\hat{\beta}^{\mathrm{BP}} - \beta\|_0 \leq \|\hat{\beta}^{\mathrm{BP}}\|_0 + \|\beta\|_0 \leq n + s$, we then have the following lower bound on $\|w^{\mathrm{BP}}\|_2$ assuming $n \geq s$,

$$\|w^{\mathrm{BP}}\|_2 \geq \frac{\|w^{\mathrm{BP}}\|_1}{\sqrt{n+s}} \geq \frac{\|w^{\mathrm{BP}}\|_1}{\sqrt{2n}}. \tag{91}$$

Combining with Prop. 36, we have proved Prop. 4.

## K.3 Tightness of the bounds on $\|w^{\mathbf{BP}}\|_1$

As we discussed in the main body of the paper, our upper and lower bounds on $\|w^{\mathrm{BP}}\|_2$ still have a significant gap. Interesting, our bounds on $\|w^{\mathrm{BP}}\|_1$ are tight up to a constant factor, which may be of independent interest. To show this, we first derive the following upper bound on $\|w^{\mathrm{BP}}\|_1$.

**Proposition 37** (upper bound on $\|w^{\mathrm{BP}}\|_1$). *When $s \leq \sqrt{\frac{n}{7168\ln(16n)}}$, if $p \in \left[(16n)^4, \exp\left(\frac{n}{1792s^2}\right)\right]$, then*

$$\|w^{BP}\|_1 \leq \left(4\sqrt{2} + \sqrt{\frac{1}{2\sqrt{7}}}\right)\sqrt{\frac{n}{\ln p}}\|\epsilon_{train}\|_2,$$

*with probability at least $1 - 6/p$.*

*Proof.* Following the proof of Theorem 2 in Appendix F, we can still get that Eq. (48), i.e.,

$$M \leq 2\sqrt{7}\sqrt{\frac{\ln p}{n}}, \quad \|w^I\|_1 \leq \sqrt{\frac{2n}{\ln p}}\|\epsilon_{\mathrm{train}}\|_2, \text{ and } K \geq 4, \tag{92}$$

hold with probability at least $1 - 6/p$. Applying Eq. (92) and Proposition 5, we have, with probability at least $1 - 6/p$,

$$\|w^{\mathrm{BP}}\|_1 \leq 4\|w^I\|_1 + \sqrt{\frac{1}{2\sqrt{7}}}\left(\frac{n}{\ln p}\right)^{1/4}\|\epsilon_{\mathrm{train}}\|_2$$

$$\leq 4\sqrt{\frac{2n}{\ln p}}\|\epsilon_{\mathrm{train}}\|_2 + \sqrt{\frac{1}{2\sqrt{7}}}\left(\frac{n}{\ln p}\right)^{1/4}\|\epsilon_{\mathrm{train}}\|_2$$

$$\leq \left(4\sqrt{2} + \sqrt{\frac{1}{2\sqrt{7}}}\right)\sqrt{\frac{n}{\ln p}}\|\epsilon_{\mathrm{train}}\|_2,$$

where the last inequality is because $\frac{n}{\ln p} > 1$, and therefore $(\frac{n}{\ln p})^{1/4} \leq (\frac{n}{\ln p})^{1/2}$. $\square$

Comparing with Prop. 36, we can see that our upper and lower bounds on $\|w^{\text{BP}}\|_1$ differ by at most a constant factor.

## K.4  Proof of Proposition 35

To prove Proposition 35, we will prove a slightly stronger result in Proposition 38 given below.

**Proposition 38.** *When $(p - s) \leq e^{(n-1)/16}/n$ and $n \geq 17$, the following holds.*

$$\frac{\|w^I\|_1}{\|\epsilon_{train}\|_2} \geq \sqrt{1 + \frac{n - 1}{4 \ln n + 4 \ln(p - s)}}, \tag{93}$$

*with probability at least $1 - 3/n$.*

To prove Proposition 38, we introduce a technical lemma first.

**Lemma 39.** *For any $x \in [0, 1)$, we have*

$$\ln(1 - x) \geq \frac{-x}{\sqrt{1 - x}}. \tag{94}$$

*Proof.* Let

$$f(x) = \ln(1 - x) + \frac{x}{\sqrt{1 - x}}.$$

Note that $f(0) = 0$. Thus, it suffices to show that $df(x)/dx \geq 0$ when $x \in [0, 1)$. Indeed, we have

$$
\begin{aligned}
\frac{df(x)}{dx} &= \frac{-1}{1 - x} + \frac{\sqrt{1 - x} - x\frac{-1}{2\sqrt{1-x}}}{1 - x} \\
&= \frac{-\sqrt{1 - x} + 1 - x + x/2}{(1 - x)^{3/2}} \\
&= \frac{2 - x - 2\sqrt{1 - x}}{2(1 - x)^{3/2}} \\
&= \frac{(1 - \sqrt{1 - x})^2}{2(1 - x)^{3/2}} \\
&\geq 0.
\end{aligned}
$$

The result of this lemma thus follows. $\qquad\square$

We are now ready to prove Proposition 38.

**Proof of Proposition 38:**  Because of Lemma 34, we only need to show that

$$\frac{\|\epsilon_{\text{train}}\|_2}{\mathbf{B}^T_{(1)}(-\epsilon_{\text{train}})} \geq \sqrt{1 + \frac{n - 1}{4 \ln n + 4 \ln(p - s)}},$$

with probability at least $1 - 3/n$. Similar to what we do in Appendix I.3, without loss of generality, we let $\epsilon_{\text{train}} = [-\|\epsilon_{\text{train}}\|_2 \; 0 \; \cdots \; 0]^T$. Thus,

$$\frac{\|\epsilon_{\text{train}}\|_2}{\mathbf{B}^T_{(1)}(-\epsilon_{\text{train}})} = \frac{1}{\max_i |\mathbf{A}_{i1}|}.$$

We uses the following two steps in order to get an upper bound of $1/\max_i |\mathbf{A}_{i1}|$. Step 1: estimate the distribution of $1/|\mathbf{A}_{i1}|$ for any $i \in \{1, \cdots, p - s\}$. Step 2: utilizing the fact that all $\mathbf{A}_{i1}$'s are independent, we estimate $1/\max_i |\mathbf{A}_{i1}|$ base on the result in Step 1.

The Step 1 proceeds as following. For any $i \in \{1, \cdots, p - s\}$ and any $k \geq 0$, we have

$$
\begin{aligned}
&\Pr\left(\left\{\frac{1}{|\mathbf{A}_{i1}|} \geq k\right\}\right) \\
&= \Pr\left(\left\{(\mathbf{A}'_{i1})^2 \leq \frac{\sum_{j=2}^n (\mathbf{A}'_{ij})^2}{k^2 - 1}\right\}\right) \text{ (by Eq. (67)).}
\end{aligned}
$$

Therefore, for any $m > 0$, we have

$$\Pr\left(\left\{\frac{1}{|\mathbf{A}_{i1}|} \ge k\right\}\right)$$

$$\ge \Pr\left(\left\{(\mathbf{A}'_{ij})^2 \le \frac{n-1-2\sqrt{(n-1)m}}{k^2-1}\right\}\right)$$

$$\cdot \Pr\left(\left\{\sum_{j=2}^{n}(\mathbf{A}'_{ij})^2 > n-1-2\sqrt{(n-1)m}\right\}\right) \quad \text{(because all } \mathbf{A}'_{ij}\text{'s are independent)}$$

$$\ge \left(1 - 2\Phi^c\left(\sqrt{\frac{n-1-2\sqrt{(n-1)m}}{k^2-1}}\right)\right)$$

$$\cdot \left(1 - e^{-m}\right) \text{ (by Lemma 11).} \tag{95}$$

Let $m = (n-1)/16$ and define

$$t := \sqrt{\frac{(n-1)/2}{k^2-1}}. \tag{96}$$

We have

$$\sqrt{\frac{n-1-2\sqrt{(n-1)m}}{k^2-1}} = t. \tag{97}$$

Substituting Eq. (97) and $m = (n-1)/16$ to Eq. (95), we have

$$\Pr\left(\left\{\frac{1}{|\mathbf{A}_{i1}|} \ge k\right\}\right) \ge \left(1 - e^{-(n-1)/16}\right)(1 - 2\Phi^c(t))$$

$$\ge \left(1 - e^{-(n-1)/16}\right)\left(1 - \frac{2\sqrt{2/\pi}e^{-t^2/2}}{t + \sqrt{t^2 + \frac{8}{\pi}}}\right) \text{ (by Lemma 24)}$$

$$\ge \left(1 - e^{-(n-1)/16}\right)\left(1 - e^{-t^2/2}\right) \text{ (since } t \ge 0 \implies t + \sqrt{t^2 + 8/\pi} \ge 2\sqrt{2/\pi}).$$

Now, let $k$ take the value of the RHS of Eq. (93), i.e.,

$$k = \sqrt{1 + \frac{n-1}{4\ln n + 4\ln(p-s)}}.$$

By Eq. (96), we have

$$t^2 = \frac{(n-1)/2}{k^2-1}$$

$$= \frac{(n-1)/2}{\left(\sqrt{1 + \frac{n-1}{4\ln n + 4\ln(p-s)}}\right)^2 - 1} \text{ (substituting the value of } k)$$

$$= 2\ln n + 2\ln(p-s),$$

which implies that

$$e^{-t^2/2} = \frac{1}{n(p-s)}.$$

Thus, we have

$$\Pr\left(\left\{\frac{1}{|\mathbf{A}_{i1}|} \ge k\right\}\right) \ge \left(1 - e^{-(n-1)/16}\right)\left(1 - \frac{1}{n(p-s)}\right). \tag{98}$$

Next, in Step 2, we use Eq. (98) to estimate $1/\max_i |\mathbf{A}_{i1}|$. Since all $\mathbf{A}_{i1}$'s are independent, we have

$$
\Pr\left(\left\{\frac{1}{\max_i |\mathbf{A}_{i1}|} \geq k\right\}\right)
$$

$$
= \prod_{i=1}^{p-s} \Pr\left(\left\{\frac{1}{|\mathbf{A}_{i1}|} \geq k\right\}\right) \quad \text{(since all } \mathbf{A}_{i1} \text{ are independent)}
$$

$$
\geq \left(\left(1 - e^{-(n-1)/16}\right)\left(1 - \frac{1}{n(p-s)}\right)\right)^{p-s} \quad \text{(by Eq. (98))}
$$

$$
= \exp\left((p-s)\ln(1 - e^{-(n-1)/16})\right)
$$

$$
\cdot \exp\left((p-s)\ln(1 - \frac{1}{n(p-s)})\right)
$$

$$
\geq \exp\left(-\frac{(p-s)e^{-(n-1)/16}}{\sqrt{1 - e^{-(n-1)/16}}}\right) \exp\left((p-s)\frac{-\frac{1}{n(p-s)}}{\sqrt{1 - \frac{1}{n(p-s)}}}\right)
$$

(by Lemma 39)

$$
= \exp\left(-\frac{(p-s)e^{-(n-1)/16}}{\sqrt{1 - e^{-(n-1)/16}}}\right) \exp\left(\frac{-1}{n\sqrt{1 - \frac{1}{n(p-s)}}}\right)
$$

$$
\geq \left(1 - \frac{(p-s)e^{-(n-1)/16}}{\sqrt{1 - e^{-(n-1)/16}}}\right)\left(1 - \frac{1}{n\sqrt{1 - \frac{1}{n(p-s)}}}\right)
$$

(because $e^x \geq 1 + x$)

$$
\geq \left(1 - \frac{1}{n\sqrt{1 - e^{-(n-1)/16}}}\right)\left(1 - \frac{1}{n\sqrt{1 - 1/17}}\right)
$$

(based on the assumption of the proposition, i.e., $p - s \leq e^{(n-1)/16}/n$ and $n(p-s) \geq n \geq 17$)

$$
\geq \left(1 - \frac{1}{n\sqrt{1 - 1/e}}\right)\left(1 - \frac{1}{n\sqrt{1 - 1/17}}\right) \quad \text{(because } n \geq 17)
$$

$$
= 1 - \frac{1}{n\sqrt{1 - 1/e}} - \frac{1}{n\sqrt{1 - 1/17}} + \frac{1}{n\sqrt{1 - 1/e}}\frac{1}{n\sqrt{1 - 1/17}}
$$

$$
\geq 1 - \frac{2}{\sqrt{1 - 1/e}} \cdot \frac{1}{n} \quad \text{(because } 17 > e)
$$

$$
\geq 1 - 3/n \quad \text{(because } e \geq 9/5).
$$

The result of this proposition thus follows. ∎

Finally, we use the following lemma to simplify the expression in Proposition 38. The result of Proposition 35 thus follows.

**Lemma 40.** *If $n \geq 17$, then*

$$
\sqrt{1 + \frac{n-1}{4\ln p + 4\ln(p-s)}} \geq \sqrt{1 + \frac{n}{9\ln p}}.
$$

*Proof.* Because $n \geq 17$, we have

$$
\frac{n-1}{n} = 1 - \frac{1}{n} \geq 1 - \frac{1}{17} \geq \frac{8}{9}.
$$

Therefore, we have

$$\frac{n-1}{n} \geq \frac{4}{9} + \frac{4}{9}$$

$$\implies \frac{n-1}{n} \geq \frac{4\ln p}{9\ln p} + \frac{4\ln(p-s)}{9\ln p}$$

$$\implies \frac{n-1}{n} \geq \frac{4\ln p + 4\ln(p-s)}{9\ln p}$$

$$\implies \frac{n-1}{4\ln p + 4\ln(p-s)} \geq \frac{n}{9\ln p}$$

$$\implies \sqrt{1 + \frac{n-1}{4\ln p + 4\ln(p-s)}} \geq \sqrt{1 + \frac{n}{9\ln p}}.$$

$\square$

## Footnotes

[4]Notice that in the proof of Proposition 5, to get Eq. (36), we do not need $K > 0$.

[5]Rotating $\epsilon_{\text{train}}$ around the origin is equivalent to rotating all columns of $\mathbf{A}$. Since the distribution of $\mathbf{A}_i$ is uniform on the unit hyper-sphere in $\mathbb{R}^n$, such rotation does not affect the objective of the problem (50).

[6]Notice that the proof of Proposition 35 does not require $s > 0$. Therefore, we can just let $s = 0$ so that $w^I$ there becomes $w^J$.