[Reviews · NeurIPS 2020]

Review 1

Summary and Contributions: The paper studies "harmless overfitting" -- is it possible to fit the training data perfectly, while generalising well to the test set? The authors aim to answer this question for a simple linear model, but unlike previous work, which focussed primarily on l2-minimising interpolants, this paper studies l1-minimising interpolants. More precisely, the setting is as follows: the data comes from a linear regression model with Gaussian inputs and additive noise; the true regression coefficient vector beta is sparse; a linear model is trained by picking the l1-norm minimiser from among solutions that fit the training data. The main result is an upper bound for the risk, in the regime where the input dimension p (which is also the size of the model) far exceeds the number of training points. The risk bound is decreasing in p, mirroring a similar "harmless overfitting" phenomenon that occurs for l2-norm minimising interpolants. However, the bound is only valid for p < exp(const.n); for larger p the risk could increase again.

Strengths: * Interesting problem, and good progress towards solving it. * Well written paper. * Nice proof idea, and very well explained too. I really appreciate the intuitive explanations of the proof ideas in Sec 4.

Weaknesses: First of all I should say that I like this paper. The following should be taken more like 'issues in need of clarification' or 'things a reader might be confused about' rather than 'weaknesses'. * The main result (Thm 2) seems a bit inconclusive, in the sense that it only holds for a range of p. It's not clear to me what happens when p exceeds the upper bracket; does the risk go up again? As far as I can tell, Thm 2 and Prop 4 don't resolve this, and I find the experimental evidence difficult to interpret (more on that below). * The gap between the upper (Thm 2) and the lower (Prop 4) bound seems quite large to me. In particular, I'd be curious if it's possible to get rid of the constant term on the RHS of (9)? Or should the constant feature in the lower bound? * I don't understand some of the interpretations of Thm 2. For example, in line 170, it says "intuitively, when the signal is strong, it should be easier to detect the true model", which I agree with, and would therefore expect the risk to decrease with ||beta||. Why doesn't it? * I find the experiments (Fig 1) confusing, maybe the authors could explain them more? First, I'd expect the risk curves (of both l1 and l2 minimisers) to be decreasing in p, isn't that what this is all about? Second, it is claimed in Section 3(i), that the risk of l1-minimisers is unaffected by the norm of beta, but there is a clear difference between the green and the orange curve (BP, beta norm = 1 or 0.1). * Related to the above, the term 'descent region' is used multiple times (e.g. in line 190) throughout the paper. It seems central to understanding the figures and the risk bounds, however, it isn't introduced or explained anywhere. For a reader who thinks of double descent as <descent-ascent-descent>, rather than <descent-ascent-descent-ascent> (which seems to be implied), this is confusing. Nitpicks: * [Sec 2] It seems to me that the distinction between scaled and unscaled coefficients complicates the setting, perhaps unnecessarily. Would it be cleaner to just assume X with normalised-Gaussian columns to start with? * In lines 127--128, it says "...even when Eq (8) has multiple solutions, there must exist one with at most n non-zero elements." This seems true geometrically, but it would be nice to have a proof. * In lines 230--231, it says "Assuming that X_train has full row-rank (...), w^I exists if and only if p - s >= n ...". Strictly speaking this isn't true, e.g. if X_train = [Identity, zero]. (But yes, w^I exists almost surely.)

Correctness: The main claims seem correct; I didn't read the proof in the Supplementary Material, but the proof sketch provided in the main text seems reasonable to me.

Clarity: Yes, the paper is well written.

Relation to Prior Work: Yes, there is discussion of prior work.

Reproducibility: Yes

Additional Feedback: ========== Post-rebuttal ============== Thank you for your response. Some of my confusion has been resolved and I'm keeping the original positive score. One (minor) point that several reviewers felt wasn't that well explained was R1.3 (in the 2D example, depending on the training point the risk could scale with ||beta||, e.g. if x=[0.5, 1] your argument doesn't work anymore). Maybe something to keep in mind for future revisions of the paper.


Review 2

Summary and Contributions: Provided an upper-bound (decreasing with the number of features) of the generalization error of the minimum $\ell_1$ norm interpolant in an non-asymptotic (and potentially overparameterized) setting. The results highlighted the difference between the minimum $\ell_1$- and $\ell_2$-norm solution in the overparameterized regime in terms of relation to the problem dimensions and the null risk.

Strengths: Understanding the generalization properties of the minimum $\ell_1$ norm interpolant is an important problem, since most of the double descent literature focuses on the least squares solution. In addition, the analysis is finite-sample and applicable beyond the proportional asymptotic limit. The rate suggested by the upper-bound is empirically supported, and the observation that more data can hurt is interesting.

Weaknesses: I find the main theorem to be unsatisfactory for the following reasons: 1. the upper-bound on the risk appears to be rather loose: it scales with number of data point and can be quite large (in fact larger than the null risk) unless p = exp(n). While this level of overparameterization is not uncommon in the compressed sensing literature, this result does not provide a complete characterization of the minimum $\ell_1$ norm interpolant in the overparameterized regime. 2. the upper-bound is only valid in a certain overparameterized region (beyond the interpolation threshold). In addition, the required sparsity also relates to the number of data points, which is a bit unnatural. 3. the unit Gaussian data assumption is a bit restrictive compared to previous works on the minimum $\ell_2$ norm interpolant. 4. showing an upper-bound of the risk that decreases with p doesn't exactly establish the double descent phenomenon (i.e. risk peaks at interpolation threshold and then decreases), especially that the considered region is way over the interpolation threshold.

Correctness: To my knowledge yes (except for previously mentioned discussion on double descent).

Clarity: Yes.

Relation to Prior Work: There is one (potentially concurrent) paper that derived the generalization error of minimum $\ell_1$ norm interpolant in the proportional asymptotics (based on CGMT): Liang, Tengyuan, and Pragya Sur. "A precise high-dimensional asymptotic theory for boosting and min-l1-norm interpolated classifiers." arXiv preprint arXiv:2002.01586 (2020). I think the difference and similarity in the finding should be discussed.

Reproducibility: Yes

Additional Feedback: I have the following questions and suggestions: 1. to what extent does the result extend beyond squared loss and isotropic Gaussian data? For instance, would it be possible to characterize the effect of eigenvalue decay, as in "Benign Overfitting in Linear Regression"? 2. the descent region specified in this work is way beyond the interpolation threshold. Would it be possible to understand the behavior of the minimum $\ell_1$ norm solution when p is around the same order as n? 3. it would be a very nice empirical confirmation if the rate of population risk does indeed follow the rate $n^{1/4}$; perhaps having a log axis for Figure 3 would make the message clearer. 4. (minor) why does the curves in Figure 2 start from different positions (x axis)? 5. (minor) while this is not the major motivation, I don't really see why DNN training relates to the minimum $\ell_1$ norm solution. ----------------------------------- Post-rebuttal: The authors addressed some of my concerns; I have therefore increased my score. A minor issue on terminology: I agree with the authors that the risk has to decrease from the peak at n=p. However, it is worth noting that such descent may not be monotonic (for the L2 case see Figure 2 in https://arxiv.org/abs/1908.10292, or Figure 7(a) in https://arxiv.org/abs/2006.05800). So "double descent" may not be the complete description for p>n.


Review 3

Summary and Contributions: minimum l_1 norm solution that perfectly fits the observed data). Most prior work focus on the l_2 norm solution, and this paper shows (1) that double descent occurs for l_1 for finite $n$, $p$, and (2) the descent curve exhibits qualitatively different phenomenon from double descent with the l_2 norm, namely: independence of the signal strength ||beta||, slower descent inversely proportional to log(p), (vs. log(p) for l_2), and a lower descent floor when the SNR is high and the signal is very sparse.

Strengths: This paper extends the recent literature on double descent to a new setting (basis pursuit/l_1 norm). In this setting, the descent phenomenon still occurs in the overparameterized interpolating regime, and there are a few new features of the descent curve that do not appear in the l_2 norm setting. It may be interesting to understand the extent to which some of these features appear in the practical settings. To my knowledge, the risk bounds for finite $n$ and $p$ in this overparameterized + interpolating setting are new, and this upper bound is accompanied by a lower bound showing the 1 / log(p) rate is inevitable. There is a small numerical example corroborating the main results of the theory.

Weaknesses: The paper fills in a gap in the double descent theory, though the motivations for studying the l_1 case may be less strong than the l_2 setting (i.e. the solution produced by SGD for linear models). Moreover, the main results work in the more restrictive setting where all of the features are i.i.d. Gaussian, though it seems likely this can be relaxed with a more involved and careful analysis. Finally, the constants in Theorem 1 are not very sharp, and the bound holds only for p in [n^4, exp(n)], as opposed to results like [Belkin 2019] for the l_2 case which work for all $p$.

Correctness: The results seem correct at a "sanity-check" level and match the empirical plots, but I did not carefully check the proofs in the appendix.

Clarity: The paper is fairly easy to read. One suggestion would be to add slightly more explanation about the numerical experiments in Figure 1 / 2. It is not clear without a careful reading of the text what the primary take-away from each figure is.

Relation to Prior Work: Yes, particularly focusing on the basis pursuit setting with finite samples $n$ and parameters $p$, $p > n$, and interpolating the training error $Y = X\beta$.

Reproducibility: Yes

Additional Feedback: ============== After authors feedback ================= Thank you for addressing my comments! I enjoyed the paper and will keep my score.


Review 4

Summary and Contributions: This paper introduces theoretical (and practical, although this part is less novel, as mentioned on line 54) evidence that the minimal L1 norm solution of overparametrized linear regression exhibits double-descent behavior, i.e. the model error reduces with increasing the number of features (and thus parameters).

Strengths: Deep learning generalizes surprisingly well, given that it can fit almost any data, which -- according to the classical learning theory -- suggests that it should generalize poorly. Attempts to bring learning theory up to date with this observation are important for further progress in deep learning, and this paper sheds light on a this question in a simplified regime: learning linear regressions models with minimizing L1 norm of the solution (instead of a more common choice -- minimizing L2 norm). The paper theoretically shows that this setup (similarly to deep learning) exhibits the so-called double descent behaviour (generalizing well even when having enough parameters to fit any dataset) for finite dataset sizes and finite number of features (in contrast to showing such results in the limit of this numbers going to infinity). This result is novel and I expect it to be of great interest to the learning theory community.

Weaknesses: There is a noticeable gap between theory and practice presented in this paper: the smallest setting where theorem 2 conditions apply and the number of non-zero parameters is `s >= 1` is n = 102581 datapoints; p = 168000000000 features; s = 1. The largest experiment result is on n = 1250; p < 1e5; (i.e. all experimental results are way outside the region where theory applies). While it’s not bad per se, I think it's worth explicitly stating this in the paper. Also, I’m not 100% sure, but it feels that when you compare the descent floor for BP and L2 solutions (lines 203 - 225), you compare both on the data generated from the ground truth BP (sparse) model, so it makes sense that the floor is better for BP. Again, it’s not bad to compare those two on the same data models, but if that’s the case I think it’s worth stating it explicitly.

Correctness: I believe that the presented results and methodology is correct (but I didn't check the proofs in the supplimentary material). However, I find it hard to believe that the model error can be independent of the signal strength (lines 160-172), can you please comment on this? To this end, I tried to experimentally verify this claim and it doesn't seem to hold in a simple experiment I did: (which doesn’t actually contradict the theorem claim, because it’s infeasible to run an experiment with problem setting large enough so that the theorem applies), on a smaller experiment (the same size as the ones done in the paper, n = 500, s = 100, p = 2000) the model error was around 6 when using beta with norm 0.1; the error was around 60 when beta norm was 1; around 600 when norm was 10; and around 6000 when norm was 100. So, it’s hard for me to believe that an upper bound on the model error may not depend on the scale of beta at all. However, please take this claim with a grain of salt: I spent just around an hour coding this experiment and the model error is surprisingly large even for smaller scale betas, so it’s very likely that I might have made some mistake. So please take this paragraph as an invitation to discussion, not as an accusation of being wrong. See here for my implementation: https://pastebin.com/xwTEuu0r

Clarity: Yes, the paper is clearly written.

Relation to Prior Work: Yes, the difference to the prior work is clearly discussed.

Reproducibility: Yes

Additional Feedback: Minor comments: Line 125 introduces \hat{\beta}^{BP} and then line 128 introduces it again with an additional constraint, which is a bit confusing. Line 136, I think there is a norm symbol missing around ||w^{BP}|| ============== After authors feedback ================= Thank you for addressing my comments and sorry for inflicting the stress of debugging my code on you. You're absolutely right, after fixing the problem my code also confirms your results. I'm happy to increase my score, well done with the paper!

[Author Response · NeurIPS 2020]

We thank all reviewers for the constructive comments. Below, we label each comment by Reviewer#.Comment#.

R1.1 [ascent when $p$ exceeds the range in Thm 2, descent-region] As $p$ becomes larger, eventually the model error
will increase again. The reason is that there will be some columns of $\mathbf{X}_{\text{train}}$ very similar to those of true features.
Then, BP will pick those very similar but wrong features, and the error will approach null risk. As a result, the whole
double-descent behavior should indeed be <descent-ascent(p<n)-descent(p>n)-ascent> (as shown in Fig. 1, the same is
also true for $\ell_2$-minimization). By the "descent region", we mean the second "descent" part. Thus, Fig. 1 shows that
model error is insensitive to $\|\beta\|_2$ before it ascends again. We will make these clear in the final version.
R1.2 [constant term on the RHS of (9)] Currently we can reduce, but cannot completely get rid of, the constant term in
(9). It does not appear in Prop. 4 due to the different path of proof (see (88) of Supplemental Material).
R1.3 [upper-bound of BP is not affected by $\|\beta\|_2$] This insensitivity to $\|\beta\|_2$ is a special feature of BP. This can be seen
by considering the following (perhaps over-simplified) example. Given one training datum ($x \in \mathbb{R}^2, y \in \mathbb{R}$) generated
by the true model $y = x^T\beta + 0.1$ (here 0.1 is the noise), where $\beta = [\beta_1, \ \beta_2]^T$, and $x = [1, \ 0.5]^T$. Assume $\beta_2 = 0$
(for sparsity). If we minimize $\|\hat{\beta}\|_1$ subject to $y = x^T\hat{\beta}$, we get $\hat{\beta} = [\beta_1 + 0.1, \ 0]^T$ (and thus the model error is always
$\|\beta - \hat{\beta}\|_2 = 0.1$) for any $\beta_1$. In a higher dimensional space, due to a similar reason (under certain conditions), the
generalization error of BP could be insensitive to $\|\beta\|_2$.

R2.1 [upper bound may be larger than the null risk] We agree. On the other hand, one important goal is to understand
in what settings min-$\ell_1$ is better than min-$\ell_2$. Eq. (9) can already provide new and useful answers, e.g., when $p$ is
exponentially large and when noise is low, the model error of min-$\ell_1$ is much lower than that of min-$\ell_2$ (null risk).
R2.2 [bounds are loose, range of $p$ does not include $p$ near $n$, squared loss and isotropic Gaussian data] We acknowledge
these limitations. We believe the results can be extended to independent non-Gaussian data, while other generalization
may require more work. However, we also would like to point out that, even under our model, the analysis for
BP is significantly more challenging than min-$\ell_2$ solutions because, unlike min-$\ell_2$ solutions that can be written as
pseudo-inverses, min-$\ell_1$ solutions have no closed form. Thus, we have to resort to large-$p$ asymptotes to capture the
model error of BP, which is why the characterization of $p \approx n$ remains unknown. We believe that this difficulty is the
reason why there are so few results on overfitting BP solutions. Despite these limitations, our results still successfully
reveal key insights on the difference between min-$\ell_1$ and min-$\ell_2$ solutions, which we believe are of significant interest
to the community (as the other reviewer commented).
R2.3 [required sparsity relates to $n$] Such requirements are not uncommon, e.g., Thm. 3.1 of [16] also requires the
sparsity to be below some function of $M$ (incoherence of $\mathbf{X}_{\text{train}}$), which is related to $n$ according to Prop. 9 in our paper.
R2.4 [establish double descent by the upper-bound] Our result already implies double descent. The reason is that, when
$p \leq n$, the MSE solution is independent of $\ell_1$ vs. $\ell_2$. From known results for min-$\ell_2$, the model error has a peak when
$p \to n$ from below. Thus, our upper bound implies that the model error of BP has to descend from that peak.
R2.5 [related work: arXiv preprint arXiv:2002.01586 (2020)] Thank you for the pointer and we will cite this reference.
However, the reference studies classification, while we study regression. The notion of "fitting the data" is quite
different between the two models. Hence, we feel that the conclusions are not directly comparable.
R2.6 [(minor) Fig. 2 starts from different positions (x axis)] That is because we focus on the interpolating regime $p > n$
and $n$ varies. We will add back the regime $p < n$ for the final version. Thank you for the suggestion!

R2.7 / R3.1 [connection to DNN, motivation of studying $\ell_1$ minimization] We agree that the connection between min-$\ell_1$
and DNN is not clear yet. Similar to other related work, our approach is to use linear models as a starting point to
understand which types of overfitting solutions approximate the generalization power of DNN better. Our hope is that
such analysis would eventually lead to better training methods than SGD (which is closely related to $\ell_2$ minimization).

R3.2 [Gaussian features assumption, limited range of $p$ compared with $\ell_2$, constants in Theorem 2 are not sharp] We
acknowledge these limitations. Please also refer to our response to R2.2.

R4.1 [$n$ and $p$ are large for theory] One reason for large $p$ and $n$ is that we aim for with-high-probability results, which
leads to stricter conditions than average-case analysis. Another reason is that we have not optimized the constants.
Nonetheless, the numerical results suggest that the main predicted trends hold for much smaller $n$ and $p$. We will clarify
this in the final version. Thank you for the suggestion!
R4.2 [model error independent of the signal strength] Thanks for sharing your code! There seems to be a small
yet consequential bug on Line 39, which should be `print(np.linalg.norm(x[:p] - beta_exact))` instead of
`print(np.linalg.norm(x[:p] - beta_exact[:, 0]))`. (The original code returns the norm of a $p \times p$ matrix
instead of a $p \times 1$ vector.) After correcting this line, the result becomes: 0.019, 0.020, 0.021, and 0.021 for $\|\beta\|_2$
equals to 0.1, 1, 10, and 100, respectively, which verifies our conclusion "model error can be independent of the signal
strength". Please also refer to our response to R1.3 for the intuition. We will post our code for the final version.

[Meta-Review · NeurIPS 2020]

The paper is a good technical contribution to a phenomenon of widespread interest to the NeurIPS community. While some reviewers had initial concerns about the somewhat strong assumptions (range of validity of parameters, Gaussianity), the technical hurdles to overcome are significant and after the discussion, they were ameliorated.